# Improving Extreme Wind Prediction with Frequency-Informed Learning

**Chenrui Xu[1], Xi Huang[2], Ying-Jun Angela Zhang[1], Jianwei Huang[3,4]** *

[1] The Chinese University of Hong Kong   [2] Around Tech (Shenzhen), Ltd.

[3] School of Science and Engineering, The Chinese University of Hong Kong, Shenzhen

[4] Shenzhen Institute of Artificial Intelligence and Robotics for Society (AIRS)

jianweihuang@cuhk.edu.cn

## Abstract

Accurate prediction of extreme wind velocities has substantial significance in industry, particularly for the operation management of wind power plants. Although the state-of-the-art data-driven models perform well for general meteorological forecasting, they may exhibit large errors for extreme weather—for example, systematically underestimating the magnitudes and short-term variation of extreme winds. To address this issue, we conduct a theoretical analysis of how the data frequency spectrum influences errors in extreme wind prediction. Based on these insights, we propose a novel loss function that incorporates a gradient penalty to mitigate the magnitude shrinkage of extreme weather, and we theoretically justify its effectiveness via a PDE-based energy–enstrophy analysis. To capture more precise short-term wind velocity variations, we design a novel structure of physics-embedded machine learning models with frequency reweighting. Experiments demonstrate that, compared to the baseline models, our approach achieves significant improvements in predicting extreme wind velocities while maintaining robust overall performance.

## 1 Introduction

Wind velocity field prediction is crucial to both academic research and industrial practice (Tascikaraoglu & Uzunoglu, 2014; Hanifi et al., 2020). For instance, the wind power plants require accurate predictions of the wind speed magnitude to support accurate real-time production estimation and safe operational control, since the output power is approximately proportional to the wind magnitude cubed ($v^3$), and the wind turbines will cease to work for extremely large wind (Sabzehgar et al., 2020; Wan et al., 2010). Traditionally, solving dynamic systems by mathematical methods, Numerical Weather Prediction (NWP) has been the workhorse of wind velocity prediction (Coiffier, 2011). However, recent advances in deep learning have revolutionized weather prediction, with models like FourCastNet (Pathak et al., 2022) and Pangu-Weather (Bi et al., 2022) significantly outperforming traditional NWP methods (Coiffier, 2011). Based on an extensive amount of data, these models are specialized in producing accurate overall predictions of the wind velocity field.

One of the key challenges in wind velocity prediction is to accurately predict the amplitude changes of the extreme wind. General data-driven models may be struggling with this challenge if they are trained by common loss functions (like MSE) based on regular wind speed datasets (Olivetti & Messori, 2024). For example, many data-driven forecasters systematically underestimate the amplitudes of extreme winds (Rychlik & Mao, 2019). This bias persists even when overall (non-extreme) skill is strong (Shi et al., 2025), leading to underestimated risks and missed rapid ramps in practical operations. Therefore, addressing this challenge in extreme wind velocity prediction is the main focus of this paper.

There exist several data-driven models specifically designed for extreme weather predictions. Some models employ classical deep learning models such as RNN (Prasetya & Djamal, 2019), CNN

---

*Corresponding author. Jianwei Huang is also affiliated with the Shenzhen Key Laboratory of Crowd Intelligence Empowered Low-Carbon Energy Network, CSIJRI Joint Research Centre on Smart Energy Storage, The Chinese University of Hong Kong, Shenzhen, and Shenzhen Loop Area Institute (SLAI), Shenzhen, China.

(Zhang et al., 2019), and LSTM (Gao et al., 2018) to capture spatiotemporal dependencies in weather data. Many other models utilize generative data augmentation methods, including variational autoencoder (VAE) (Vega-Bayo et al., 2024) and diffusion models (Zhong et al., 2024), to address data scarcity through weather pattern simulation. Despite these advances, critical gaps remain for extreme wind predictions: (i) most approaches offer little theoretical (or even intuitive) explanation of why errors arise significantly for extreme winds; (ii) many methods implicitly rely on abundant training data that include extreme cases, whereas such cases are intrinsically scarce in real datasets; (iii) to better capture the dynamics of sharply-changing pattern, the data-driven models may require much more complicated model structure than overall prediction; and (iv) depending mainly on large data, some models may be insufficient for capture the intrinsic dynamics and still suffer from uncontrollable regional errors for extreme weather prediction (Zhou et al., 2024).

To resolve the challenge, we conduct a theoretical analysis of the error behavior in the frequency domain. Based on proper simplification, we separate the traditional mean-squared error (MSE) into three terms: amplitude shrinkage error, pattern translation error, and noise. We show that while training with standard MSE as a loss function, small pattern deviations will lead to significant amplitude shrinkage for the high-frequency wind field components, causing the underestimation of extreme amplitude and blurred short-term variability in prediction. Inspired by the analysis, we propose a gradient-penalized loss function upweighting the amplitude shrinkage error. We further provide a PDE-based energy–enstrophy interpretation showing that the new loss function enforces enstrophy matching and controls small-scale vorticity, thereby theoretically explaining its effectiveness in mitigating amplitude shrinkage for extreme winds. To more effectively reduce the pattern translation error, as well as improve parameter efficiency, we design a physics-embedded structure for the neural network, with a backbone of the Navier-Stokes (NS) equation. The equations reveal how the motion of a fluid, such as the atmosphere, is affected by a combination of external forces, pressures within the fluid, and viscous effects (Marion & Temam, 1998). Moreover, we utilize a frequency separation and reweighting mechanism to coordinate the impact of high- and low-frequency components to the loss function. Based on the above frequency-informed modification of the loss function and neural network structure, our model overcomes the amplitude shrinkage challenge in wind prediction and achieves a significant improvement in extreme wind velocity prediction accuracy. Our research makes the following significant contributions to the field of extreme wind prediction:

- **Frequency-theoretic explanation of underestimation.** We provide Fourier-domain analyses showing how small spatial shifts and scaling yield a frequency-dependent MSE, theoretically accounting for underestimation of extreme amplitudes of short-term variability.

- **Gradient-penalized objective for extremes.** We propose a simple, implementation-ready loss that augments MSE with gradient matching, equivalently reweighting high-frequency errors to mitigate spectral shrinkage and recover sharp ramps.

- **Frequency separation & reweighting with a physics-embedded backbone.** We design a spectral pipeline (Fourier masking, Frequency separation) atop a simplified NS block, targeting precise short-horizon dynamics while preserving stability and data efficiency.

- **Empirical validation on regional extremes.** Across diverse regions and baselines (including CNN, ConvLSTM, and PINN), our method substantially improves extreme-wind prediction while maintaining robust overall performance under normal conditions.

The remainder of this paper is structured as follows: Section 2 includes problem formulation, physical backgrounds and theoretical insights. Section 3 displays our methodology, including the novel loss function formulation and network design. Section 4 describes our experiments and corresponding results. Section 5 contains the conclusion with limitations and future directions. Related works to this paper can be found in Appendix A. Other details are provided in the appendices.

## 2 PRELIMINARIES AND INSIGHTS

### 2.1 PROBLEM FORMULATION

In this paper, we mainly consider the wind velocity field prediction within certain rectangular regions, which can be discretized into $N \times M$ points. Let $\mathbf{u}(\mathbf{x}, t) = [v(x, y, t), w(x, y, t)]$ denote the wind velocity field in this region, where $\mathbf{x} = (x, y)$ is the two-dimensional spatial do-

main and $t$ is the temporal domain; $v$ and $w$ represent the velocity components in the longitude and latitude directions, respectively. We denote historical wind data sequences by $\mathbf{u}_{[t_1:t_N]} = \{\mathbf{u}(x,y,t_1), \mathbf{u}(x,y,t_2), \ldots, \mathbf{u}(x,y,t_N)\}$.

Let $\tilde{\mathbf{u}}(\mathbf{x},t)$ denote the prediction of $\mathbf{u}$ given by a certain model at time $t$. Then our objective to predict the wind velocity field at the next time can be expressed as follows:

$$\tilde{\mathbf{u}}_{t_{N+1}} = \tilde{\mathbf{u}}(x,y,t_{N+1}) = f_\theta(\mathbf{u}_{[t_1:t_N]}, \text{other data}), \tag{1}$$

where $f_\theta$ is the model we intend to train, and "other data" contains other data sequences that might also contribute to wind velocity prediction (like surface pressure, which will be explained later). The error between $\mathbf{u}$ and $\tilde{\mathbf{u}}$ evaluates the performance of the predictive model.

**Temporal and Spatial Scale for Extreme Prediction.** In atmospheric forecasting, temporal and spatial scales are tightly coupled: short-term predictions are typically associated with short-range dynamics (Jung & Broadwater, 2014) (Zhu et al., 2019). In this paper, we adopt the convention of extreme wind velocity prediction that focuses on short-period and regional prediction, while the temporal and spatial resolutions are also higher compared to global weather forecasting to resolve rapidly evolving, small-scale features.

## 2.2 Physical Background

As a fundamental assumption in meteorology, atmospheric systems' dynamics generally satisfy the *Navier-Stokes (NS) equations* together with the *continuity equation* (mass-conservation) constraint(Holton & Hakim, 2013). The NS equations are a set of nonlinear partial differential equations (PDEs) that describe the relationship between the motion of a fluid and the forces acting upon it. For a two-dimensional domain with wind velocity field $\mathbf{u} = [v(x,y,t), w(x,y,t)]$, the NS equation and continuity equation are shown as follows:

$$\begin{cases} \frac{\partial \mathbf{u}}{\partial t} = -\mathbf{u}\cdot\nabla\mathbf{u} - \frac{1}{\rho}\nabla P + \nu\nabla^2\mathbf{u} + \mathbf{F} & x \in \Omega,\ t > 0, \\ \nabla\cdot\mathbf{u} = 0, & x \in \Omega,\ t > 0, \end{cases} \tag{2}$$

where $\mathbf{u}\cdot\nabla\mathbf{u}$ is the advective acceleration; $-\frac{1}{\rho}\nabla P$ refers to the pressure gradient force; $\nu\nabla^2\mathbf{u}$ denotes the viscous friction ($\nu$: kinematic viscosity); and $\mathbf{F}$ is the external body forces. The body force term may vary in different scenarios, with typical examples including gravity and the Coriolis force (Holton & Hakim, 2013).

## 2.3 Insights from Frequency Domain Analysis

When solving and analyzing PDEs, a standard method is to apply the Fourier transform (often with respect to spatial domains) to convert the PDEs into ODEs (Evans, 2022). Let $\hat{\cdot}$ denote the Fourier operator, and $\mathbf{k} = (k_x, k_y)$ denote the frequency domain coordinates. For example, applying the Fourier transform to a simplified version of the NS equation 2 with advection and diffusion:

$$\partial_t \mathbf{u}(\mathbf{x},t) + \mathbf{U}\cdot\nabla\mathbf{u}(\mathbf{x},t) = \nu\nabla^2\mathbf{u}(\mathbf{x},t) + f(\mathbf{x},t),$$

we will get

$$\partial_t \hat{\mathbf{u}}(\mathbf{k},t) + i\,(\mathbf{k}\cdot\mathbf{U})\,\hat{\mathbf{u}}(\mathbf{k},t) = -\nu\|\mathbf{k}\|^2\hat{\mathbf{u}}(\mathbf{k},t) + \hat{f}(\mathbf{k},t), \tag{3}$$

which is an ODE with respect to $\hat{\mathbf{u}}$. The equation 3 shows that the advection of air may appear as a phase shift $e^{-i\,\mathbf{k}\cdot\mathbf{U}\,t}$, corresponding to a spatial translation in physical space. Moreover, diffusion may induce amplitude damping at a rate proportional to $\|\mathbf{k}\|^2$ (stronger for high frequency).

Motivated by this idea, we apply a 2-dimensional Fourier transform to the wind velocity fields and analyze how the frequency spectrum affects the prediction error. By equation 3 and statistical convention, we assume that the prediction error is mainly caused by three factors: scaling, translation, and noise. Therefore, the relationship between $\tilde{\mathbf{u}}$ and $\mathbf{u}$ can be illustrated as follows:

$$\tilde{\mathbf{u}}(\mathbf{x}) = a\mathbf{u}(\mathbf{x} + \Delta) + \varepsilon(\mathbf{x}), \tag{4}$$

where $a$ is the scaling magnitude of wind speed amplitude, and $\Delta$ corresponds to the deviation amount of the data pattern. We may also assume $\varepsilon \sim \mathcal{N}(0, \sigma^2)$ to be a Gaussian noise.

Now, let's consider the mean squared error (MSE) of the prediction and the ground-truth. By Rayleigh's energy theorem (Temple et al., 2004), we can show that the MSE of the original data in the spatial domain is equivalent to the MSE of the Fourier-transformed data in the (double) frequency domain:

$$\text{MSE}(\mathbf{u}, \tilde{\mathbf{u}}) = \frac{1}{NM} \sum_{\mathbf{x}} |\mathbf{u} - \tilde{\mathbf{u}}|^2 = \frac{1}{(NM)^2} \sum_{k} \left| \hat{\mathbf{u}} - \hat{\tilde{\mathbf{u}}} \right|^2.$$

We denote $\theta_{\mathbf{k}} = 2\pi \left( \frac{k_x \Delta_x}{N} + \frac{k_y \Delta_y}{M} \right)$, and we assume $\theta_k$ is sufficiently small. Then the Fourier transform of the prediction is $\hat{\tilde{\mathbf{u}}}(k) = a e^{i\theta_k} \hat{\mathbf{u}}(k) + \varepsilon(k)$, and the expectation of the MSE will be:

$$
\begin{aligned}
\mathbb{E}[\text{MSE}(\mathbf{u}, \tilde{\mathbf{u}})] &= C_1 \sum_{\mathbf{k}} \left( 1 - a e^{i\theta_{\mathbf{k}}} \right)^2 ||\hat{\mathbf{u}}(\mathbf{k})||^2 + \sigma^2; \\
&= C_1 \sum_{\mathbf{k}} \left( a^2 + 1 - 2a \mathbb{E}[\cos \theta_{\mathbf{k}}] \right) \cdot ||\hat{\mathbf{u}}(\mathbf{k})||^2 + \sigma^2; \\
&= C_1 \sum_{\mathbf{k}} \underbrace{\{a - \mathbb{E}[\cos \theta_{\mathbf{k}}]\}^2 ||\hat{\mathbf{u}}(\mathbf{k})||^2}_{\text{scaling error}} + \underbrace{\{1 - \mathbb{E}^2[\cos \theta_{\mathbf{k}}]\} ||\hat{\mathbf{u}}(\mathbf{k})||^2}_{\text{translation error}} + \underbrace{\sigma^2}_{\text{noise}},
\end{aligned}
\tag{5}
$$

where $C_1$ is a constant depending on $N$ and $M$.

As the last line of equation 5 shows, the MSE has been separated into three components: The *scaling error* reflects the magnitude difference between the prediction and the ground-truth; the *translation error* is caused by the pattern deviation $\Delta$; the *noise* is assumed to be independent of both $a$ and $\Delta$.

**The Cause of Amplitude Shrinkage.** The scaling error term is highly related to the pattern deviation factor $\Delta$, and the theoretically optimal amplitude scaling will be $a = \mathbb{E}[\cos \theta_{\mathbf{k}}]$. Given the existence of the deviation $\Delta$, we will have $\mathbb{E}[\cos \theta_{\mathbf{k}}] < 1$, causing the shrinkage in the amplitude of the predicted wind speed. Therefore, we also name the *scaling error* as *shrinkage error*.

Moreover, if we assume that $\Delta$ is small enough, then the optimal $a$ will be:

$$a_{opt} = \mathbb{E}[\cos \theta_{\mathbf{k}}] = 1 - \frac{C_2 (\mathbf{k} \cdot \Delta)^2}{2} + o(||\mathbf{k}||^2), \tag{6}$$

which is decreasing as $\mathbf{k}$ becomes higher and $C_2$ is a scalar. Therefore, the amplitude shrinkage phenomenon will theoretically tend to be more severe for high-frequency spectral components.

When trained with MSE, the estimator reduces the squared discrepancy between the prediction and the target, effectively acting on a decomposition of error into translation, scaling, and stochastic noise. If model capacity or optimization is insufficient to avoid the translation component ($\Delta > 0$), gradients can still decrease the objective by attenuating the field's amplitude (i.e., driving $a < 1$). This mechanism explains why general MSE-trained models may underestimate wind-speed amplitudes and dampen short-term variability, thereby degrading performance on extreme-wind prediction. These results yield three practical insights for improving extreme-wind prediction:

- **Upweight scaling error.** Increase the relative weight of the amplitude (scaling) component in the loss to counteract shrinkage.
- **Reduce translation error.** Incorporate mechanisms that explicitly address misalignment $\Delta$ so the optimizer need not compensate by damping amplitudes.
- **Frequency-aware weighting.** Reweight residuals by frequency spectra to mitigate high-frequency attenuation and preserve short-term variability.

## 3 METHODOLOGY

Guided by the insights from Section 2 on frequency-domain error behavior, we propose a new *gradient-penalized* loss function that mitigates MSE-induced amplitude shrinkage under pattern deviation, and we design a neural framework that combines a *physics-embedded structure* and *frequency separation & reweighting*. The model architecture is shown in Figure 1.

## 3.1 Gradient-Penalized Loss Function

Building on the previous analysis in Section 2.3, the amplitude shrinkage phenomena under MSE mainly arise from pattern deviation between the predicted and true wind fields. One idea to solve the problem is to modify MSE by a correction term, which should be insensitive to such deviation, but capture the field's general spatial change. One of the intuitive approaches is encouraging the norm of the prediction gradient $\|\nabla\tilde{\mathbf{u}}\|$ to match that of the ground-truth $\|\nabla\mathbf{u}\|$. Therefore, we propose our novel *Gradient-Penalized Loss Function* as follows:

$$\mathcal{L}_{\mathrm{gp}}(\tilde{\mathbf{u}}, \mathbf{u}) = \mathrm{MSE}(\tilde{\mathbf{u}}, \mathbf{u}) + \lambda \left| \|\nabla\tilde{\mathbf{u}}\|^2 - \|\nabla\mathbf{u}\|^2 \right|. \tag{7}$$

The coefficient $\lambda > 0$ balances pointwise fit against global variation matching: a larger $\lambda$ more strongly discourages amplitude shrinkage and preserves high-frequency variability; a smaller $\lambda$ approaches plain MSE.

**Connection with error decomposition.** Due to the amplitude shrinkage phenomena studied in Section 2.3, $\|\nabla\tilde{\mathbf{u}}\|^2$ is likely less than $\|\nabla\mathbf{u}\|^2$ in practice. When this happens, minimizing equation 7 is equivalent to minimizing the following simplified version:

$$\mathcal{L}_{\mathrm{gp}}(\tilde{\mathbf{u}}, \mathbf{u}) = \mathrm{MSE}(\tilde{\mathbf{u}}, \mathbf{u}) - \lambda\|\nabla\tilde{\mathbf{u}}\|^2. \tag{8}$$

Applying the Fourier transform to $\nabla\tilde{\mathbf{u}}$ and using the Rayleigh's energy theorem as in Section 2.3, we know $\|\nabla\tilde{\mathbf{u}}\|^2$ is proportional to $\sum_{\mathbf{k}} \|\mathbf{k}\|^2 \left\|\hat{\tilde{u}}(\mathbf{k})\right\|^2$. On the other hand, suppose that $\Delta$ is sufficiently small, then the decomposition equation 5 will also yield

$$\mathbb{E}[\mathrm{MSE}] \approx C_1 \sum_{\mathbf{k}} (a-1)^2\|\hat{\mathbf{u}}(\mathbf{k})\|^2 + C_2^2\, a\, \cos\langle\mathbf{k}, \Delta\rangle\|\Delta\|^2\|\mathbf{k}\|^2 \|\hat{\mathbf{u}}(\mathbf{k})\|^2 + \sigma^2,$$

where the second term is proportional to $\sum_{\mathbf{k}} \cos\langle\mathbf{k}, \Delta\rangle \frac{\|\Delta\|^2}{a}\|\mathbf{k}\|^2 \|\hat{\tilde{u}}(\mathbf{k})\|^2$, and thus proportional to $\|\nabla\tilde{\mathbf{u}}\|^2$. Therefore, the essential effect of the gradient-penalization can be explained as follows: By tuning $\lambda$, the loss $\mathcal{L}_{\mathrm{gp}}$ increases the effective weight on shrinkage error relative to pattern translation error. Consequently, when optimization hits a bottleneck in reducing the translation mismatch, the model will attempt to optimize on the shrinkage error and thus improve the extreme prediction.

## 3.2 Energy–Enstrophy Interpretation of the Gradient-Penalized Loss

The motivation of our gradient-penalized loss function 7 is strongly connected to the energy structure of the incompressible NS equations (Leslie & Shvydkoy, 2016). Conceptually, the loss function can be viewed as trading off *energy matching* (via the MSE term) and *enstrophy matching* (via the gradient term) (Dascaliuc et al., 2005; Palha & Gerritsma, 2017). This perspective explains why gradient penalization is theoretically effective in mitigating amplitude shrinkage of high-gradient, extreme wind events. More detailed explanations and proofs to all theorems can be found in Appendix C.2.

In fluid dynamics, vorticity and enstrophy are often used to characterize the rotation and turbulent behavior of a fluid. The formal definition of them is shown as follows (Foias et al., 2001).

**Definition 1** (Vorticity). *For an incompressible velocity field* $\mathbf{u}(u_1, u_2) : \Omega \to \mathbb{R}^2$, *the vorticity* $\omega$ *is defined to be the curl of the fluid velocity, mathematically,*

$$\omega(x, t) := (\nabla \times \mathbf{u})(x, t) = \partial_{x_1} u_2(x, t) - \partial_{x_2} u_1(x, t).$$

**Definition 2** (Enstrophy). *The enstrophy* $\mathcal{E}$ *of the flow* $\mathbf{u}$ *is defined as the* $L^2$*–norm of the vorticity,*

$$\mathcal{E}(\mathbf{u}(t)) := \int_{\Omega} |\omega(x, t)|^2 \, \mathrm{d}x = \int_{\Omega} |\nabla \times \mathbf{u}(x, t)|^2 \, \mathrm{d}x. \tag{9}$$

According to Foias et al. (2001), for incompressible NS equations, there exist an important connection between enstrophy and the norm of the velocity field, illustrated by the following theorem.

**Theorem 1** (Enstrophy of 2D incompressible flow). *Let* $\mathbf{u}$ *be a sufficiently regular, divergence-free 2D velocity field on* $\Omega$ *with periodic or no-slip boundary conditions. Then its enstrophy is equivalent to the* $L^2$ *norm of the velocity gradient:*

$$\mathcal{E}(\mathbf{u}) \simeq \int_{\Omega} \|\nabla\mathbf{u}(x)\|^2 \, \mathrm{d}x = \|\nabla\mathbf{u}\|_{L^2}^2. \tag{10}$$

Thus, $\|\nabla\mathbf{u}\|_{L^2}^2$ can be interpreted as the total strength of rotation and shear in the flow: it is dominated by fronts, shear layers, and small-scale vortices where $\|\nabla\mathbf{u}\|$ is large.

**Energy balance for 2D incompressible NS equations.** Denote the kinetic energy of velocity field $\mathbf{u}$ by $E(t) := \frac{1}{2}\|\mathbf{u}(t)\|_{L^2}^2$. Taking the $L^2$ inner product of NS equation 2 with $\mathbf{u}(t)$ and integrating by parts (see Appendix C.2), we obtain the following basic energy balance theorem.

**Theorem 2** (Energy balance for incompressible NS). *Let $\mathbf{u}$ be a sufficiently regular solution of equation 2 with periodic or no-slip boundary conditions. Then, for almost every $t \geq 0$, the kinetic energy satisfies*

$$\frac{1}{2}\frac{\mathrm{d}}{\mathrm{d}t}\|\mathbf{u}(t)\|_{L^2}^2 + \nu\|\nabla\mathbf{u}(t)\|_{L^2}^2 = \langle \mathbf{F}(t), \mathbf{u}(t)\rangle. \tag{11}$$

This theorem shows that the enstrophy $\|\nabla\mathbf{u}(t)\|_{L^2}^2$ controls the rate at which kinetic energy is dissipated by viscosity, and therefore plays a central role in the evolution of kinetic energy.

**Spectral representation of energy and enstrophy.** Let $E(k, t)$ be the spectral energy at frequency $k$. By Rayleigh's theorem (see appendix B), we have

$$\begin{aligned}
E(t) &= \frac{1}{2}\|\mathbf{u}(t)\|_{L^2}^2 = \frac{1}{2}\int_0^\infty E(k, t)\,\mathrm{d}k, \\
\mathcal{E}(t) &= \int_\Omega \|\nabla\mathbf{u}(x, t)\|^2\,\mathrm{d}x = \int_0^\infty k^2 E(k, t)\,\mathrm{d}k.
\end{aligned} \tag{12}$$

Thus, the same spectrum $E(k, t)$ generates both the kinetic energy (an unweighted integral) and the enstrophy (a $k^2$-weighted integral) (Fischer et al., 2007). In particular, enstrophy is heavily biased toward high frequencies (small scales), while energy treats all scales equally.

**Gradient-penalized loss as an energy–enstrophy trade-off.** Based on the above analysis, the continuous form of the gradient-penalized loss function 7 can be interpreted as follows:

$$\mathcal{L}_{\mathrm{gp}}(\tilde{\mathbf{u}}, \mathbf{u}) = \underbrace{\int_\Omega \|\tilde{\mathbf{u}}(x) - \mathbf{u}(x)\|^2\,\mathrm{d}x}_{\text{energy matching (MSE)}} + \lambda \underbrace{\left\lfloor \|\nabla\tilde{\mathbf{u}}\|_{L^2}^2 - \|\nabla\mathbf{u}\|_{L^2}^2 \right\rfloor}_{\text{enstrophy matching}}, \tag{13}$$

where the "energy matching" here means reducing the energy of the error field.

Here, the first term minimizes the kinetic energy of the error field, while the second term matches the enstrophy, which controls the temporal decay rate of kinetic energy. Consequently, the gradient-penalized loss does not only suppress the error energy, but also constrains the predicted flow to exhibit a physically consistent rate of energy change. In practice, the learned model thus maintains an overall accurate prediction of the velocity field, while restoring sufficient gradient and vorticity strength in high-impact regions so that the total enstrophy remains comparable to the ground truth. Therefore, the gradient-penalized loss makes uniform amplitude shrinkage an inefficient way to reduce the objective, encouraging the network to preserve the magnitude of physically relevant small-scale structures.

### 3.3 PHYSICS-EMBEDDED STRUCTURE

To more effectively reduce the *translation error* highlighted at the end of Section 2.3, we introduce a physics-embedded structure that leverages the NS equations as inductive bias. The translation error predominantly stems from uncertainty in the direction and magnitude of the wind-field shift at the next time step. Traditional neural networks do not impose explicit constraints on such pattern transport: they attempt to learn it implicitly via the loss. In contrast, using a physics-embedded backbone provides a first-principles estimate of the dominant transport and deformation of the field (e.g., advection and diffusion), yielding a rough but informative pattern forecast. This explicit physical guidance both constrains translation error more directly and reduces the burden on the learnable components, thereby lowering parameter and training costs.

Inspired by equation 2, we embed the Navier-Stokes equation into our neural network and name it as *NS Operator*. The operator is decomposed into four components:

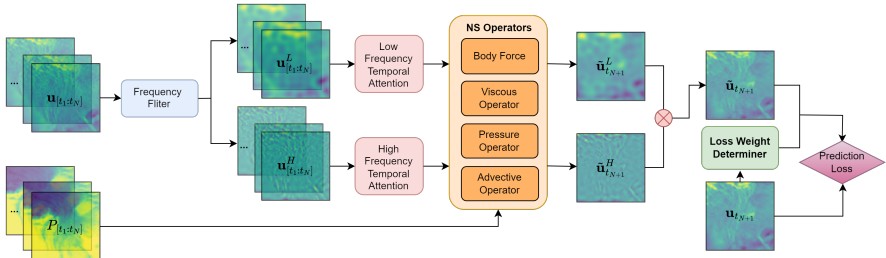

Figure 1: The full architecture of our model. The input data are the wind velocity field $\mathbf{u}$ and the pressure field $P$. The wind velocity field will be successively processed by the Frequency filter, temporal attention module, and the NS operator to obtain the prediction of high- and low-frequency data. Then the model will combine the two predictions to produce the final prediction results.

1. **Advective Operator**: implements the nonlinear transport term $\mathbf{u} \cdot \nabla \mathbf{u}$.

2. **Viscous Operator**: implements viscous diffusion $\nu \nabla^2 \mathbf{u}$ arising from internal friction.

3. **Pressure Operator**: accounts for the pressure-gradient force $\frac{1}{\rho} \nabla P$. Here, we will utilize the pressure data $P[t_1 : t_N]$. However, if the pressure data is not obtainable, we may consider the pressure force as implicit and merge this operator into the Body-Force operator.

4. **Body-Force Operator** (conventional neural networks): because explicit short-term formulations of external forces are often imprecise or unavailable, we model the body force with learnable neural networks that capture dynamics not explained by the above three operators.

Note that the **body force operator** is equivalent to conventional neural networks and can adopt various structures as other pure data-driven models (but likely contains fewer layers and parameters). During training, the first three operators will learn to generate a rough pre-prediction based on dynamic properties and to ensure the pattern translation magnitude and direction lie in a reasonable range, while the Body-Force operator will learn how to capture the exact wind field dynamics based on the pre-prediction given by the other three operators.

## 3.4 FREQUENCY DOMAIN SEPARATION AND REWEIGHTING

Guided by the insights from Section 2.3, we adopt a *frequency-aware weighting* strategy to counter high-frequency attenuation and preserve short-term variability. Concretely, we design a frequency filtering & reweighting scheme that (i) splits the wind field into low- and high-frequency components and (ii) processes and reweights these components respectively so the model can retain rapid, localized dynamics without sacrificing large-scale coherence.

**Fourier Filter.** We employ a Fourier filter (Alleyne & Cawley, 1991) (Münch et al., 2009) to decompose wind velocity data $\mathbf{u}$ into low-frequency ($\mathbf{u}^L$) and high-frequency ($\mathbf{u}^H$) components. The filter consists of three main steps: 1) Fourier Transform: Converts wind velocity data from the positional domain to frequency domain. 2) Frequency Masking: Separates high- and low-frequency components using appropriate masks: $\hat{\mathbf{u}}_f(k) = \hat{\mathbf{u}}(k) \cdot \mathcal{M}(k)$, where $\mathcal{M}(k)$ denotes the frequency mask (high or low). 3) Inverse Fourier Transform: Transforms the filtered components back to the positional domain. This decomposition enables the model to focus on distinct frequency components, enhancing its ability to capture both large-scale trends and rapid, localized variations.

**Frequency-Based Temporal Attention.** To refine the dynamic modeling, we design temporal attention mechanisms for both high- and low-frequency data sequences. Inspired by SENet (Cheng et al., 2016), temporal attention contains two operations: **Squeeze**, which compresses the data of each time slot into a value; and **Excitation**, which produces weight sequences that reflect the relative importance of each time slot for future predictions. The temporal attention is applied at different resolutions for high- and low-frequency components, respectively. Since high-frequency data are more critical for short-term dynamics, they are processed with higher temporal resolution (shorter time intervals). Conversely, low-frequency sequences, which correspond to long-term trends, are

handled at lower temporal resolution. This differentiation ensures that the model effectively captures the unique characteristics of both short-term and long-term dynamics.

# 4 EXPERIMENTAL RESULTS

We evaluated our approaches through three key experiments: 1. **Effect of Gradient-Penalized Loss Function**: Our novel loss function effectively resolves the amplitude shrinkage problem in extreme wind prediction. 2. **Main Prediction Results**: Our model outperformed baselines in both overall accuracy and predictions in extreme wind regions. 3. **Different Frequency Masking Levels**: The results showed that intermediate masking thresholds achieved the best balance between high- and low-frequency information, leading to more accurate predictions.

**Data.** We evaluate our approaches on meteorological data sampled from the 5th generation of the ECMWF reanalysis (ERA5) database (Hersbach et al., 2020). The dataset includes three key meteorological variables related to wind prediction: the eastward and northward components of 10-meter wind and surface pressure. Guided by the prediction scales stated in Section 2.1, the data of each variable is represented as a time series of two-dimensional latitude–longitude fields over the study region, temporally ordered and co-registered on a common grid. The temporal resolution of the data is 1 hour, while the spatial resolution is $0.25°$. For convenience, we define each 24-hour period as a prediction unit, where the first 23 hours are used as inputs to predict the 24th hour.

**Baseline Models.** To study the effect of the gradient-penalized loss function, we utilize the structure of a multivariate meteorological data fusion wind prediction network called MFWPN (Zhang et al., 2025). We further compare our full model with several state-of-the-art regional weather prediction approaches, including CNN, Convolutional LSTM (Tan et al., 2023), and Physics-Informed Neural Network (PINN) (Eivazi et al., 2022). We remark that the PINN model is designed with a revised form of the Navier-Stokes (NS) equations (the Reynolds-averaged Navier-Stokes (RANS) equations) (Ling & Templeton, 2015; Cai & Wang, 2024).

**Evaluation Metrics.** We assess the performance of models using *Root Mean Squared Error (RMSE)*, one of the most commonly used metrics for overall predictions. We also evaluate the *Extreme Attentive RMSE (Ex-RMSE)*, which is a modified version of RMSE focusing on regions with extreme wind velocities. The detailed explanations of the two metrics are shown in Appendix E.2.

## 4.1 EFFECT OF GRADIENT-PENALIZED LOSS FUNCTION

To quantify the impact of the proposed gradient-penalized objective function, we first compare the performance of models trained by equation 7 and MSE over the same baseline structure, and study the impact of different hyperparameter $\lambda$ on the model performance. The baseline model here adopts the same structure as MFWPN (Zhang et al., 2025), which is a machine learning model for short-term wind speed prediction using spatial-temporal fusion and CNN units. We use second-order central differences along both axes to represent spatial gradients. All other settings (optimizer, learning rate, augmentations, and early stopping criteria) are kept identical to the baseline for a fair comparison. The choices of hyperparameters include $\lambda \in \{0, 0.01, 0.02, 0.03, 0.05, 0.07, 0.10, 0.15, 0.20, 0.25\}$.

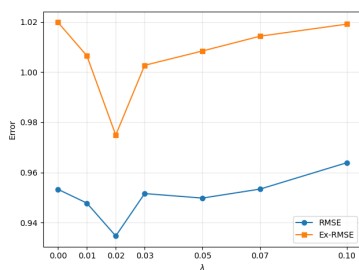

Figure 2: Effect of the gradient-penalized loss across $\lambda$ values.

**Results.** Figure 2 demonstrates the effectiveness of the gradient-penalized loss function and the trade-off between amplitude error and translation error. With a proper $\lambda$ value, the models trained by gradient-penalized loss outperform the baseline trained by MSE in general. We also observe a consistent U-shaped curve w.r.t. $\lambda$: small positive values markedly reduce extreme attentive error while preserving overall accuracy; too-large values overweight high-frequency residuals and harm stability. In particular, the best performance is achieved

Table 1: Comparative error results across models. The best values are shown in **bold**.

| Model | Lead time: 1h | | Lead time: 3h | | Lead time: 5h | |
|---|---|---|---|---|---|---|
| | RMSE | Ex-RMSE | RMSE | Ex-RMSE | RMSE | Ex-RMSE |
| CNN | 0.4639 | 0.3183 | 1.0442 | 0.7355 | 2.0757 | 1.0693 |
| ConvLSTM | 0.3471 | 0.2294 | 0.7834 | 0.5357 | 1.0644 | 0.8097 |
| PINN | 0.3946 | 0.2541 | 0.8283 | 0.5646 | 1.1434 | 0.7347 |
| **Ours** | **0.3287** | **0.1868** | **0.6622** | **0.4329** | **0.9076** | **0.6158** |

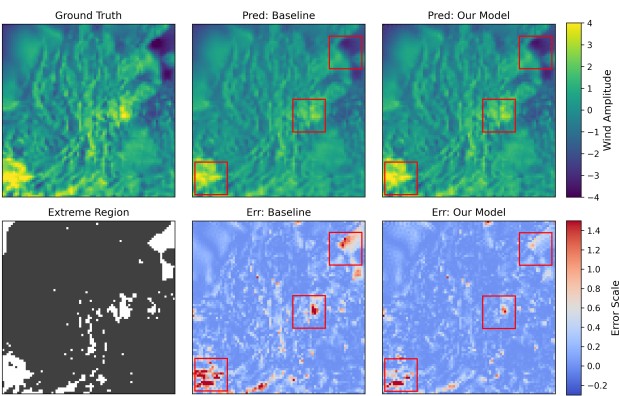

Figure 3: The first line: the ground truth and prediction results of the baseline (PINN) and our models. The first sub-figure in the second line: regions where wind velocity exceeds a specified threshold, highlighted as "Extreme Regions" (in white). The last two sub-figures: comparative prediction errors between our model and the baseline, where bluer indicate lower prediction errors.

at $\lambda = \lambda^\star$, which is 0.02 in this experiment. Moreover, when $\lambda \geq 0.15$, optimization becomes unstable and the models fail to converge within the prescribed training budget.

The empirical trend aligns with our frequency-domain analysis in Section 2.3 and Section 3.1. The gradient term in equation 7 effectively penalizes the amplitude shrinkage trend and therefore improves the accuracy for extreme wind velocity prediction. However, the penalized term $|\,\|\nabla\tilde{\mathbf{u}}\|^2 - \|\nabla\mathbf{u}\|^2\,|$ itself does not contain any information regarding positional alignment. Therefore, beyond a threshold of $\lambda$, the learned model may tend to generate predictions with large spatial fluctuations, regardless of the positional pattern mismatch. Moreover, we also provide a convergence analysis for the gradient-penalized objective in Appendix D, which theoretically explains why the model performance will degrade for large $\lambda$.

## 4.2 EXTREME WIND VELOCITY PREDICTION

In this section, we evaluate the performance of our final model, which integrates all components in the methodology section with an architecture shown in Figure 1. We compare against several representative baselines on the same regional wind velocity prediction task. The baseline models include: CNN, ConvLSTM, and PINN. We train all models with an initial learning rate of $1 \times 10^{-5}$ and the SGD optimizer. All approaches consume the same wind-velocity inputs, except that our model additionally utilizes surface pressure $P$ as an auxiliary field.

**Results.** Table 1 reports both overall prediction errors and extreme wind attentive errors. Compared to the CNN, ConvLSTM, and PINN baselines, in next-frame prediction, the overall RMSEs of our model decrease by 29.1%, 5.3%, and 16.7%, respectively; while the extreme attentive RMSEs decrease by 41.3%, 18.6%, and 26.5%. Moreover, we evaluate the performance for multi-horizon prediction. Our method consistently attains the lowest RMSE and Ex-RMSE across all longer forecasting lead times (3h and 5h). In weather forecasting, lead time is the period between issuing a forecast and the occurrence of the predicted weather (Easterling & Mjelde, 1987). Overall, these

results demonstrate that the proposed model delivers stable multi-horizon improvements and substantially enhances extreme wind prediction without sacrificing overall accuracy.

Figure 3 provides a visual comparison of regional wind velocity amplitudes. Compared with the PINN baseline, our predictions exhibit larger and more realistic amplitudes that are closer to the ground truth, particularly in the most extreme zones (highlighted by red boxes). This aligns with our frequency-domain analysis in Section 2.3 and the first-stage results in Section 4.1. The above results show our model's better performance for both extreme and overall wind velocity predictions, mitigating the critical amplitude shrinkage problem in extreme weather prediction.

### 4.3 ABLATION STUDY

To provide insights into the effectiveness of the main modules in our model, we conduct an ablation study to quantify their contributions. We compare the performance of our final model with the following cases: only NS operator (NS op); without gradient-penalized loss (W/O grad-loss); without NS structure, or in other words, only body force operator (W/O NS); and without frequency separation (W/O freq-sep). The results of the ablation study are shown in Table 2.

When we retain only the NS operator, the errors become substantially larger, indicating that simply learning a neural NS operator alone is insuf-

Table 2: Ablation study of the proposed model.

| Model | RMSE | Ex-RMSE |
|---|---|---|
| NS op | 0.7061 | 0.4577 |
| W/O grad-loss | 0.3351 | 0.2632 |
| W/O NS | 0.3754 | 0.2363 |
| W/O freq-sep | 0.4199 | 0.2703 |
| Ours | 0.3287 | 0.1868 |

ficient for the predictions. Removing the gradient-penalized term leads to only a negligible change in the global RMSE, but causes a pronounced degradation in Ex-RMSE, confirming that the gradient penalty has a targeted effect on reconstructing sharp gradients and vorticity in high-impact regions and thus sharpening extreme-wind predictions. When the NS-based structure is removed, both RMSE and Ex-RMSE deteriorate, suggesting that encoding the NS prior into the model architecture provides a beneficial inductive bias that improves the background velocity field. Moreover, if the frequency separation mechanism is discarded, the model yields a significant increase in both RMSE and Ex-RMSE, highlighting that explicitly disentangling low- and high-frequency components is crucial for capturing both large- and small-scale structures of the velocity fields.

Overall, these results demonstrate that all the above components (the NS-informed architecture, the gradient-penalized loss, and the frequency separation module) make complementary and nontrivial contributions for our full model.

## 5 CONCLUSIONS

We developed a frequency-informed learning framework to address a key obstacle in wind velocity prediction: the amplitude misalignment (particularly underestimation) of extreme wind velocity prediction. From a frequency-domain perspective, we showed that small spatial pattern deviations combined with standard MSE training induce frequency-dependent amplitude shrinkage, disproportionately suppressing high-frequency components. Guided by this insight, we separated the error into different components and proposed a gradient-penalized loss function that encourages models to emphasize amplitude misalignment. Furthermore, we provided a PDE-based energy–enstrophy analysis to explain the proposed loss function's effectiveness in restoring small-scale gradients and vorticity. We proposed a frequency separation and reweighting framework with a physics-embedded backbone to further enhance the capture of extreme wind dynamics. Empirically, the proposed methods outperform baselines on regional datasets, significantly improving extreme-wind prediction accuracy while preserving robust overall wind prediction performance.

**Limitations.** Our analysis relies on simplified assumptions about the factors (scaling, shifting, and noise) that cause prediction errors, and a comprehensive study on more complex error-causing factors might be a promising direction. Moreover, generalizations to longer lead times, 3-dimensional scenarios, and other weather variables may also be interesting.

ACKNOWLEDGMENTS

This paper is supported by Longgang District Shenzhen's "Ten Action Plan" for Supporting Innovation Projects under Grant LGKCSDPT 2024004.

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

## A    RELATED WORKS

### A.1    NUMERICAL AND DATA-DRIVEN WEATHER PREDICTION

Numerical Weather Prediction (NWP) (Bauer et al., 2015) represents traditional physical weather forecasting methods, which rely on computational techniques to solve the physical equations governing atmospheric dynamics. For wind velocity prediction, the Navier-Stokes equations and the continuity equation are pivotal in describing the wind velocity field dynamics within a region. NWP models discretize these equations over a computational grid and solve them numerically using methods such as finite difference, finite volume, or spectral techniques. Despite their widespread use, NWP models face significant limitations, including a reliance on precise initialization data and high computational costs. These challenges make real-time predictions and extreme weather scenario forecasting particularly difficult.

In contrast, purely data-driven models leverage machine learning algorithms to predict wind speed by identifying patterns in historical data. Examples include CNNs (Liu et al., 2016), LSTMs (Yu et al., 2019), ConvLSTMs (Kim et al., 2017), GANs (Li et al., 2021), and transformers (Bi et al., 2022). These models excel at capturing complex wind patterns and local variations, demonstrating flexibility and adaptability in learning from data. However, they lack the physical constraints required to ensure realistic predictions, which can sometimes result in unreasonable outputs.

### A.2    PHYSICS-INFORMED DATA-DRIVEN WEATHER PREDICTION

Recent advancements in weather prediction have introduced hybrid approaches that integrate physical laws with machine learning. For example, Physics-Informed Neural Networks (PINNs) (Cai et al., 2021; Cai & Wang, 2024) incorporate differential equations into the training process to enforce physical realism. These methods reduce dependency on large datasets and computational resources, ensure predictions adhere to known physical laws, and enhance robustness in complex environments.

A notable example of this approach is DeepPhysiNet (Li et al., 2024), developed by W. Li et al. This model combines physics-guided machine learning with weather prediction by constructing physics networks based on multilayer perceptrons for meteorological variables. Partial Differential Equations are incorporated as part of the loss function, while a hyper-network based on deep learning directly learns weather patterns, contributing to the weights of the physics networks. This hybrid design ensures both physical consistency and the ability to capture intricate weather patterns.

### A.3    EXTREME WEATHER PREDICTION

In this paper, extreme weather refers to the outlying values of specific weather properties. For example, wind power plants require precise predictions of wind speeds at turbine locations, particularly the extreme values of wind speeds (Sabzehgar et al., 2020). According to (Wan et al., 2010), when wind speeds exceed the cut-out speed, wind turbines cease operation to prevent damage. As such, accurate regional wind speed forecasting, especially for extreme wind speeds, is critical for optimizing wind power generation.

Recent advancements in machine learning have significantly improved predictive capabilities for extreme weather conditions. For instance, Fuxi Extreme, developed by X. Zhong et al. (Zhong et al., 2024), leverages a Denoising Diffusion Probabilistic Model (DDPM) to enhance accuracy and detail in extreme weather predictions. This model combines a base weather prediction framework with DDPM, capturing fine-scale features through a two-step process: adding noise in a forward step and refining details in a reverse denoising step. This innovative approach has demonstrated exceptional accuracy and detail restoration, making it highly effective for forecasting extreme weather conditions.

## B    ANALYSIS ON THE IMPACT OF FREQUENCY ON LOSS FUNCTION

In this section, we study how the frequency of the Fourier-transformed data affects the MSE loss function. Suppose we discretize the 2-dimensional region by $N \times M$ points (corresponding to

longitudinal and latitudinal directions, respectively). Then the discrete Fourier transform of the wind velocity field $\mathbf{u}$ on the region is given by:

$$\hat{\mathbf{u}}(k_x, k_y) = \mathcal{F}(\mathbf{u}) = \sum_{x=0}^{N-1} \sum_{y=0}^{M-1} \mathbf{u}(x, y) \cdot e^{-2\pi i \left( \frac{k_x x}{N} + \frac{k_y y}{M} \right)},$$

where $\hat{\cdot}$ is the Fourier operator, $k = (k_x, k_y)$ are the mode indices in the frequency domain. For simplicity, we let $\phi_k(x, y) = e^{-2\pi i \left( \frac{k_x x}{N} + \frac{k_y y}{M} \right)}$ denote the Fourier basis. Then the inverse transform can be written by:

$$\mathbf{u}(x, y) = \frac{1}{NM} \sum_k \hat{\mathbf{u}}(k) \phi_k.$$

Suppose we have a prediction of $\mathbf{u}$ denoted by $\tilde{\mathbf{u}}$. The mean square error (MSE) of the prediction is given by $\text{MSE}(\mathbf{u}, \tilde{\mathbf{u}}) = \frac{1}{NM} \sum_{x,y} |\mathbf{u} - \tilde{\mathbf{u}}|^2$ By the discrete orthogonality of the Fourier basis $\phi_k$'s, we have $\sum_{x,y} \phi_k \overline{\phi_{k'}} = NM \delta_{k,k'}$. Therefore, we can derive the equivalent form of the MSE in the frequency domain:

$$\begin{aligned}
\text{MSE}(\mathbf{u}, \tilde{\mathbf{u}}) =& \frac{1}{NM} \sum_{x,y} |\mathbf{u} - \tilde{\mathbf{u}}|^2 \\
=& \frac{1}{NM} \sum_{x,y} \left| \frac{1}{NM} \sum_k \left( \hat{\mathbf{u}}(k) - \hat{\tilde{\mathbf{u}}}(k) \right) \phi_k(x, y) \right|^2 \\
=& \frac{1}{NM} \sum_{x,y} \frac{1}{(NM)^2} \sum_k \sum_{k'} \left( \hat{\mathbf{u}}(k) - \hat{\tilde{\mathbf{u}}}(k) \right) \cdot \overline{\left( \hat{\mathbf{u}}(k') - \hat{\tilde{\mathbf{u}}}(k') \right)} \phi_k(x, y) \overline{\phi_{k'}(x, y)} \\
=& \frac{1}{(NM)^3} \sum_k \sum_{k'} \left( \hat{\mathbf{u}}(k) - \hat{\tilde{\mathbf{u}}}(k) \right) \cdot \overline{\left( \hat{\mathbf{u}}(k') - \hat{\tilde{\mathbf{u}}}(k') \right)} \underbrace{\sum_{x,y} \phi_k(x, y) \overline{\phi_{k'}(x, y)}}_{NM \delta_{k,k'}} \\
=& \frac{1}{(NM)^2} \sum_k \left| \hat{\mathbf{u}} - \hat{\tilde{\mathbf{u}}} \right|^2.
\end{aligned}$$

Now, let's analyze the potential impact of frequency on MSE. Statistically, the prediction of the wind velocity field can be modeled as a translation of the ground truth plus white noise:

$$\tilde{\mathbf{u}}(\mathbf{x}) = \mathbf{u}(\mathbf{x} + \Delta) + \varepsilon(\mathbf{x}),$$

where $\mathbf{x} = (x, y)$; $\Delta = (\Delta_x, \Delta_y)$; $\varepsilon(\mathbf{x}) \sim \mathcal{N}(0, \sigma_{\mathbf{x}}^2)$ is a Gaussian white noise that may or may not be invariant with respect to $\mathbf{x}$. It should be clarified that the real prediction scenario can be much more complicated than this formula. However, it will be extremely difficult or even impossible to study the real cases in detail. Besides, this simplification is enough to show some insights about the impact of frequency on the loss function. We denote $\theta_k = 2\pi \left( \frac{k_x \Delta_x}{N} + \frac{k_y \Delta_y}{M} \right)$, then the Fourier transform of the prediction is:

$$\hat{\tilde{\mathbf{u}}}(k) = e^{i\theta_k} \hat{\mathbf{u}}(k) + \varepsilon(k).$$

Therefore, the mean square error of the prediction is:

$$\text{MSE}(\mathbf{u}, \tilde{\mathbf{u}}) = \frac{1}{(NM)^2} \sum_k \left| (1 - e^{i\theta_k}) \hat{\mathbf{u}}(k) - \hat{\varepsilon}(k) \right|^2.$$

Since $\varepsilon$ is Gaussian white noise, we have $\mathbb{E}[\hat{\varepsilon}] = 0$ and $\mathbb{E}[\hat{\varepsilon}(k) \overline{\hat{\varepsilon}(k')}] = NM\sigma^2 \delta_{k,k'}$. Therefore, the expectation of the MSE will be:

$$\begin{aligned}
\mathbb{E}[\text{MSE}(\mathbf{u}, \tilde{\mathbf{u}})] =& \frac{1}{(NM)^2} \sum_k \left| (1 - e^{i\theta_k}) \right|^2 |\hat{\mathbf{u}}(k)|^2 + \sigma^2. \\
=& \frac{1}{(NM)^2} \sum_k 4 \sin^2(\frac{\theta_k}{2}) \cdot |\hat{\mathbf{u}}(k)|^2 + \sigma^2.
\end{aligned}$$

Suppose the magnitude of the translation $|\Delta|$ is sufficiently small such that $\theta_k \ll 1$. Then we can approximate the $\sin(\theta_k/2)$ term by $\theta/2$:

$$\mathbb{E}[\text{MSE}] \approx \frac{1}{(NM)^2} \sum_k \theta_k^2 \, |\hat{\mathbf{u}}(k)|^2 + \sigma^2$$

$$\approx C \sum_k |k|^2 \cdot |\hat{\mathbf{u}}|^2 + \sigma^2,$$

where $C$ is constant.

According to the above formula, we can separate the MSE into two parts: one is attributed to the translation of the overall wind velocity field, and the other is attributed to the essential error from the white noise. The error from translation is approximately proportional to the square of the magnitude of the frequency.

The above analysis provides us with insights that the high-frequency data is more likely to affect the MSE.

## C  PHYSICAL INTERPRETATION OF THE GRADIENT-PENALIZED LOSS: ENERGY-ENSTROPHY TRADE-OFF

### C.1  ENSTROPHY AND ITS RELATION TO THE SQUARED GRADIENT NORM

In fluid dynamics, vorticity and enstrophy are often used to characterize and quantify the rotation and turbulent behavior of a fluid. The former definitions of them can be found in Definition 1 and 2. Since the book (Foias et al., 2001) didn't provide the full justification of the Theorem 1, here we prove it in details.

*Proof.* We first express $|\omega|^2$ in terms of the components of $\nabla \mathbf{u}$. By definition,

$$\omega = \partial_{x_1} u_2 - \partial_{x_2} u_1,$$

so that

$$|\omega|^2 = (\partial_{x_1} u_2 - \partial_{x_2} u_1)^2$$
$$= (\partial_{x_1} u_2)^2 + (\partial_{x_2} u_1)^2 - 2 \, \partial_{x_1} u_2 \, \partial_{x_2} u_1.$$

On the other hand, the squared Frobenius norm of the gradient is

$$\|\nabla \mathbf{u}\|_{\text{F}}^2 = \sum_{i,j=1}^{2} \left( \partial_{x_j} u_i \right)^2 = (\partial_{x_1} u_1)^2 + (\partial_{x_2} u_1)^2 + (\partial_{x_1} u_2)^2 + (\partial_{x_2} u_2)^2.$$

A short algebraic rearrangement shows that,

$$|\omega|^2 = \sum_{i,j=1}^{2} \left( \partial_{x_j} u_i \right)^2 - \sum_{i,j=1}^{2} \partial_{x_i} u_j \, \partial_{x_j} u_i. \tag{14}$$

(Indeed, expanding the right-hand side and cancelling terms yields exactly the expression for $|\omega|^2$ above.)

Integrating equation 14 over $\Omega$ gives

$$\int_\Omega |\omega|^2 \, \mathrm{d}x = \int_\Omega \sum_{i,j=1}^{2} \left( \partial_{x_j} u_i \right)^2 \mathrm{d}x - \int_\Omega \sum_{i,j=1}^{2} \partial_{x_i} u_j \, \partial_{x_j} u_i \, \mathrm{d}x. \tag{15}$$

We now show that the second integral on the right-hand side vanishes by integration by parts and incompressibility. For clear notation we denote

$$I := \int_\Omega \sum_{i,j=1}^{2} \partial_{x_i} u_j \, \partial_{x_j} u_i \, \mathrm{d}x.$$

Using integration by parts in the $x_i$–variable and the assumption that boundary terms vanish, we obtain

$$I = \sum_{i,j=1}^{2} \int_{\Omega} \partial_{x_i} u_j \, \partial_{x_j} u_i \, \mathrm{d}x$$

$$= -\sum_{i,j=1}^{2} \int_{\Omega} u_j \, \partial_{x_i} \partial_{x_j} u_i \, \mathrm{d}x.$$

Since partial derivatives commute, $\partial_{x_i}\partial_{x_j} u_i = \partial_{x_j}\partial_{x_i} u_i$, we may rewrite this as

$$I = -\sum_{j=1}^{2} \int_{\Omega} u_j \, \partial_{x_j} \Big( \sum_{i=1}^{2} \partial_{x_i} u_i \Big) \, \mathrm{d}x.$$

By incompressibility, $\sum_{i=1}^{2} \partial_{x_i} u_i = \nabla \cdot \mathbf{u} = 0$, and hence $\partial_{x_j}(\nabla \cdot \mathbf{u}) = 0$ for $j = 1, 2$. It follows that the integrand in the last expression vanishes identically, and therefore

$$I = 0.$$

Returning to equation 15, we conclude that

$$\int_{\Omega} |\omega|^2 \, \mathrm{d}x = \int_{\Omega} \sum_{i,j=1}^{2} \left( \partial_{x_j} u_i \right)^2 \mathrm{d}x = \int_{\Omega} \|\nabla \mathbf{u}\|_{\mathrm{F}}^2 \, \mathrm{d}x,$$

which is precisely equation 10. This shows that, for 2D incompressible flows under the stated boundary conditions, the enstrophy defined via the vorticity coincides with the gradient energy of the velocity field. □

## C.2 ENERGY BALANCING FOR THE 2D INCOMPRESSIBLE NAVIER–STOKES EQUATIONS

We consider the incompressible Navier–Stokes equations on a bounded domain $\Omega \subset \mathbb{R}^2$, with viscosity $\nu > 0$ and density $\rho > 0$. As a convention in PDE analysis, we combine the NS equation with the continuity equation, which stands for the incompressibility of the fluid system. We have

$$\begin{cases} \frac{\partial \mathbf{u}}{\partial t} = -\mathbf{u} \cdot \nabla \mathbf{u} - \frac{1}{\rho}\nabla P + \nu\nabla^2 \mathbf{u} + \mathbf{F} & x \in \Omega, \ t > 0, \\ \nabla \cdot \mathbf{u} = 0, & x \in \Omega, \ t > 0, \end{cases} \tag{16}$$

where $\mathbf{u}(x,t) \in \mathbb{R}^2$ is the velocity field, $P(x,t) \in \mathbb{R}$ is the pressure, and $\mathbf{F}(x,t) \in \mathbb{R}^2$ is an external forcing (in our context, represented by the neural network). We may also assume either periodic boundary conditions or no–slip boundary conditions $\mathbf{u}|_{\partial\Omega} = 0$, so that all boundary terms arising from integration by parts vanish.

We denote the $L^2$ inner product and norm as follows:

$$\langle \mathbf{a}, \mathbf{b} \rangle := \int_{\Omega} \mathbf{a}(x) \cdot \mathbf{b}(x) \, \mathrm{d}x, \qquad \|\mathbf{u}\|_2^2 := \int_{\Omega} |\mathbf{u}(x)|^2 \, \mathrm{d}x.$$

Following the classical energy method dealing with PDEs, we take the inner product of equation 16 with $\mathbf{u}$ over $\Omega$, which yields

$$\langle \partial_t \mathbf{u}, \mathbf{u} \rangle = -\langle (\mathbf{u} \cdot \nabla)\mathbf{u}, \mathbf{u} \rangle - \frac{1}{\rho}\langle \nabla P, \mathbf{u} \rangle + \nu\langle \nabla^2 \mathbf{u}, \mathbf{u} \rangle + \langle \mathbf{F}, \mathbf{u} \rangle. \tag{17}$$

We now treat each term in equation 17 separately.

**Time derivative term.** Assuming sufficient regularity to interchange differentiation and integration, we have

$$\|\mathbf{u}(t)\|_2^2 = \int_{\Omega} |\mathbf{u}(x,t)|^2 \, \mathrm{d}x,$$

and thus

$$\frac{\mathrm{d}}{\mathrm{d}t}\|\mathbf{u}(t)\|_2^2 = \int_{\Omega} \partial_t(|\mathbf{u}|^2) \, \mathrm{d}x = \int_{\Omega} 2\,\mathbf{u} \cdot \partial_t \mathbf{u} \, \mathrm{d}x = 2\langle \partial_t \mathbf{u}, \mathbf{u} \rangle.$$

Hence

$$\langle \partial_t \mathbf{u}, \mathbf{u} \rangle = \frac{1}{2}\frac{\mathrm{d}}{\mathrm{d}t}\|\mathbf{u}(t)\|_2^2. \tag{18}$$

**Advection term.** Write $\mathbf{u} = (u_1, u_2)$ and note that
$$(\mathbf{u} \cdot \nabla)\mathbf{u} = \big(u_1 \partial_{x_1} u_1 + u_2 \partial_{x_2} u_1, \ u_1 \partial_{x_1} u_2 + u_2 \partial_{x_2} u_2\big).$$

Then
$$\langle (\mathbf{u} \cdot \nabla)\mathbf{u}, \mathbf{u} \rangle = \int_\Omega (\mathbf{u} \cdot \nabla)\mathbf{u} \cdot \mathbf{u} \, dx$$
$$= \int_\Omega \sum_{i=1}^2 (\mathbf{u} \cdot \nabla) u_i \, u_i \, dx = \int_\Omega \sum_{i=1}^2 \sum_{j=1}^2 u_j (\partial_{x_j} u_i) u_i \, dx.$$

Notice that $u_i \partial_{x_j} u_i = \frac{1}{2} \partial_{x_j}(u_i^2)$, so
$$\sum_{i=1}^2 u_i \partial_{x_j} u_i = \frac{1}{2} \partial_{x_j} \Big( \sum_{i=1}^2 u_i^2 \Big) = \frac{1}{2} \partial_{x_j} |\mathbf{u}|^2.$$

Therefore,
$$\langle (\mathbf{u} \cdot \nabla)\mathbf{u}, \mathbf{u} \rangle = \int_\Omega \sum_{j=1}^2 u_j \cdot \frac{1}{2} \partial_{x_j} |\mathbf{u}|^2 \, dx$$
$$= \frac{1}{2} \int_\Omega \mathbf{u} \cdot \nabla(|\mathbf{u}|^2) \, dx.$$

Using the identity
$$\mathbf{u} \cdot \nabla(|\mathbf{u}|^2) = \nabla \cdot \big( \mathbf{u}|\mathbf{u}|^2 \big) - |\mathbf{u}|^2 (\nabla \cdot \mathbf{u}),$$

we obtain
$$\langle (\mathbf{u} \cdot \nabla)\mathbf{u}, \mathbf{u} \rangle = \frac{1}{2} \int_\Omega \nabla \cdot \big( \mathbf{u}|\mathbf{u}|^2 \big) \, dx - \frac{1}{2} \int_\Omega |\mathbf{u}|^2 (\nabla \cdot \mathbf{u}) \, dx.$$

Since $\nabla \cdot \mathbf{u} = 0$, the second term vanishes. Applying the divergence theorem to the first term gives
$$\int_\Omega \nabla \cdot \big( \mathbf{u}|\mathbf{u}|^2 \big) \, dx = \int_{\partial\Omega} |\mathbf{u}|^2 (\mathbf{u} \cdot \mathbf{n}) \, dS,$$

where $\mathbf{n}$ is the outward unit normal on $\partial\Omega$. Under periodic boundary conditions, this boundary integral cancels out; under no–slip conditions, $\mathbf{u} = 0$ on $\partial\Omega$ so the integrand vanishes. In either case,
$$\langle (\mathbf{u} \cdot \nabla)\mathbf{u}, \mathbf{u} \rangle = 0. \tag{19}$$
Thus the convection term does not contribute to the evolution of the energy.

**Viscous term.** We now consider the viscous term
$$\nu \langle \nabla^2 \mathbf{u}, \mathbf{u} \rangle = \nu \int_\Omega \nabla^2 \mathbf{u} \cdot \mathbf{u} \, dx$$
$$= \nu \sum_{i=1}^2 \int_\Omega \nabla^2 u_i \, u_i \, dx.$$

Since $\nabla^2 u_i = \nabla \cdot (\nabla u_i)$, integration by parts yields
$$\int_\Omega \nabla^2 u_i \, u_i \, dx = \int_\Omega \nabla \cdot (\nabla u_i) \, u_i \, dx = \int_{\partial\Omega} (\partial_n u_i) u_i \, dS - \int_\Omega |\nabla u_i|^2 \, dx.$$

Therefore,
$$\nu \langle \nabla^2 \mathbf{u}, \mathbf{u} \rangle = \nu \sum_{i=1}^2 \left[ \int_{\partial\Omega} (\partial_n u_i) u_i \, dS - \int_\Omega |\nabla u_i|^2 \, dx \right]$$
$$= -\nu \sum_{i=1}^2 \int_\Omega |\nabla u_i|^2 \, dx + \nu \sum_{i=1}^2 \int_{\partial\Omega} (\partial_n u_i) u_i \, dS.$$

As before, under periodic boundary conditions the boundary integral vanishes by periodicity; under no–slip conditions $u_i = 0$ on $\partial\Omega$, so the boundary term is again zero. Thus
$$\nu \langle \nabla^2 \mathbf{u}, \mathbf{u} \rangle = -\nu \sum_{i=1}^2 \int_\Omega |\nabla u_i|^2 \, dx = -\nu \int_\Omega |\nabla \mathbf{u}|^2 \, dx = -\nu \|\nabla \mathbf{u}\|_2^2. \tag{20}$$
This term represents viscous dissipation of kinetic energy.

**Pressure term.** For the pressure term we use integration by parts:

$$\frac{1}{\rho}\langle\nabla P, \mathbf{u}\rangle = \frac{1}{\rho}\int_\Omega \nabla P \cdot \mathbf{u}\,\mathrm{d}x = -\frac{1}{\rho}\int_\Omega P\left(\nabla \cdot \mathbf{u}\right)\mathrm{d}x + \frac{1}{\rho}\int_{\partial\Omega} P(\mathbf{u} \cdot \mathbf{n})\,\mathrm{d}S.$$

Since $\nabla \cdot \mathbf{u} = 0$, the volume integral vanishes. Under periodic boundary conditions, the boundary integral cancels; under no–slip conditions $\mathbf{u} = 0$ on $\partial\Omega$, so $\mathbf{u} \cdot \mathbf{n} = 0$ and the boundary integral is zero. Consequently,

$$\frac{1}{\rho}\langle\nabla P, \mathbf{u}\rangle = 0. \tag{21}$$

Substituting equation 18–equation 21 into equation 17 yields the following theorem.

**Theorem 3** (Energy balance for NS equation). *Let $\Omega \subset \mathbb{R}^2$ be a bounded domain with smooth boundary, $\mathbf{u}$ be a velocity field defined on $\Omega$ satisfying the incompressible Navier-Stokes equation 16 with corresponding boundary conditions. Then the velocity field $\mathbf{u}$ satisfies the* energy balance equality

$$\frac{1}{2}\frac{\mathrm{d}}{\mathrm{d}t}\|\mathbf{u}(t)\|_{L^2(\Omega)}^2 + \nu\|\nabla\mathbf{u}(t)\|_{L^2(\Omega)}^2 = \langle\mathbf{F}(t), \mathbf{u}(t)\rangle. \tag{22}$$

*for all $t$ in the interval of existence.*

**Physical interpretation.** The equation 22 can be interpreted as an exact balance of kinetic energy (per unit density) for the incompressible flow:

$$\underbrace{\frac{1}{2}\frac{\mathrm{d}}{\mathrm{d}t}\|\mathbf{u}(t)\|_{L^2}^2}_{\text{rate of change of kinetic energy}} + \underbrace{\nu\|\nabla\mathbf{u}(t)\|_{L^2}^2}_{\text{viscous dissipation}} = \underbrace{\langle\mathbf{F}(t), \mathbf{u}(t)\rangle}_{\text{power input by forcing}}.$$

The first term measures how fast the total kinetic energy $E(t) = \frac{1}{2}\|\mathbf{u}(t)\|_{L^2}^2$ changes in time. The second term is nonnegative and represents viscous dissipation: it is proportional to $\|\nabla\mathbf{u}\|_{L^2}^2$, which in two dimensions is equivalent (up to constants) to the enstrophy $\|\nabla\times\mathbf{u}\|_{L^2}^2$ and therefore quantifies the strength of small-scale shear and vortical structures. The right-hand side $\langle\mathbf{F}, \mathbf{u}\rangle$ is the rate at which the external forcing $\mathbf{F}$ does work on the flow, injecting kinetic energy into the system. In our setting, $\mathbf{F}$ is represented by the neural network and thus encodes the part of the dynamics that is not explicitly captured by the analytic Navier–Stokes operator.

## C.3 Frequency-Domain Perspectives of Energy and Enstrophy

We now consider the energy balance in frequency domain. This makes explicit how kinetic energy and enstrophy are distributed across spatial scales, and will later be used to interpret the proposed gradient-penalized loss.

Let $E(t)$ and $\mathcal{E}(t)$ denote the total kinetic energy and enstrophy, respectively.

$$E(t) := \frac{1}{2}\|\mathbf{u}(t)\|_{L^2(\Omega)}^2, \qquad \mathcal{E}(t) := \|\nabla\mathbf{u}(t)\|_{L^2(\Omega)}^2.$$

According to Rayleigh's theorem (Temple et al., 2004) illustrated in appendix B, these quantities can be written in terms of the Fourier coefficients of $\mathbf{u}$:

$$E(t) = \frac{1}{2}\sum_{k\in\mathbb{Z}^2}|\widehat{\mathbf{u}}(k,t)|^2, \qquad \mathcal{E}(t) = \sum_{k\in\mathbb{Z}^2}|k|^2\,|\widehat{\mathbf{u}}(k,t)|^2.$$

For isotropic flows, it is convenient to introduce a one–dimensional energy spectrum $E(k,t)$ such that

$$E(t) = \frac{1}{2}\int_0^\infty E(k,t)\,\mathrm{d}k, \qquad \mathcal{E}(t) = \int_0^\infty k^2 E(k,t)\,\mathrm{d}k. \tag{23}$$

Given this representation, we may consider the energy and enstrophy as follows: they are both yielded by the same spectra $\{E(k,t)\}_k$; the energy is an unweighted integral, while the enstrophy includes a $k^2$ weight that emphasizes high frequency.

Analogizing the analysis of theorem 3 to spectral energy, we obtain the following theorem in the frequency domain.

**Theorem 4** (Frequency-domain energy balance). *Let $\Omega \subset \mathbb{R}^2$ be a bounded domain with smooth boundary, $\mathbf{u}$ be a velocity field defined on $\Omega$ satisfying the incompressible Navier-Stokes equation 16 with corresponding boundary conditions. Let $E(k,t)$ be the isotropic energy spectrum defined above, and let $G(k,t)$ denote the rate of energy injection by $\mathbf{F}$ at frequency $k$. Then the total kinetic energy $E(t)$ satisfies the frequency-domain energy balance equation*

$$\frac{\mathrm{d}}{\mathrm{d}t}E(t) = -2\nu \int_0^\infty k^2 E(k,t)\,\mathrm{d}k + \int_0^\infty G(k,t)\,\mathrm{d}k. \tag{24}$$

*Proof.* Taking the spatial Fourier transform of the Navier–Stokes equations yields

$$\partial_t \widehat{\mathbf{u}}(k,t) = -\nu |k|^2 \widehat{\mathbf{u}}(k,t) + \widehat{\mathcal{N}}(k,t) + \widehat{\mathbf{F}}(k,t), \tag{25}$$

where $\widehat{\mathcal{N}}(k,t)$ collects the contributions of the advection and pressure terms. Because the flow is incompressible, the pressure term acts as an orthogonal projection onto divergence-free modes and does not contribute directly to the energy balance.

Taking the inner product in $\mathbb{C}^2$ of equation 25 with $\widehat{\mathbf{u}}(k,t)$ and then keeping the real part (denoted as $\Re(\cdot)$), we obtain a spectrum-wise energy balance:

$$\frac{1}{2}\frac{\mathrm{d}}{\mathrm{d}t}|\widehat{\mathbf{u}}(k,t)|^2 = \Re\big(\partial_t \widehat{\mathbf{u}}(k,t) \cdot \overline{\widehat{\mathbf{u}}(k,t)}\big)$$
$$= -\nu |k|^2 |\widehat{\mathbf{u}}(k,t)|^2 + \Re\big(\widehat{\mathcal{N}}(k,t) \cdot \overline{\widehat{\mathbf{u}}(k,t)}\big) + \Re\big(\widehat{\mathbf{F}}(k,t) \cdot \overline{\widehat{\mathbf{u}}(k,t)}\big). \tag{26}$$

The nonlinear term $\widehat{\mathcal{N}}(k,t)$ redistributes energy among spectra. However, when summed over all $k$, its contribution cancels due to the skew-symmetry of the advection operator, which is the spectral counterpart of $\langle (\mathbf{u} \cdot \nabla)\mathbf{u}, \mathbf{u} \rangle = 0$ in physical space. Therefore,

$$\sum_k \Re\big(\widehat{\mathcal{N}}(k,t) \cdot \overline{\widehat{\mathbf{u}}(k,t)}\big) = 0.$$

Summing equation 26 over $k$ and using Rayleigh's energy theorem to identify $E(t) = \frac{1}{2}\sum_k |\widehat{\mathbf{u}}(k,t)|^2$ and $\mathcal{E}(t) = \sum_k |k|^2 |\widehat{\mathbf{u}}(k,t)|^2$, we obtain

$$\frac{\mathrm{d}}{\mathrm{d}t}E(t) = -2\nu \mathcal{E}(t) + \sum_{k \in \mathbb{Z}^2} \Re\big(\widehat{\mathbf{F}}(k,t) \cdot \overline{\widehat{\mathbf{u}}(k,t)}\big).$$

Passing from the discrete sum over $k \in \mathbb{Z}^2$ to the isotropic $E(k,t)$ and power injection density $G(k,t)$ yields equation 24, with

$$\mathcal{E}(t) = \int_0^\infty k^2 E(k,t)\,\mathrm{d}k, \qquad \int_0^\infty G(k,t)\,\mathrm{d}k = \sum_{k \in \mathbb{Z}^2} \Re\big(\widehat{\mathbf{F}}(k,t) \cdot \overline{\widehat{\mathbf{u}}(k,t)}\big).$$

This completes the proof. $\qquad\qquad\qquad\qquad\qquad\qquad\qquad\qquad\qquad\qquad\qquad\qquad\square$

**Special case without external force.** The simple unforced case $\mathbf{F} \equiv 0$ already offers an intuitive viewpoint on why, data-driven predictors are particularly prone to amplitude shrinkage when a flow's energy is heavily concentrated in high-frequency spectra, as implied by equation 6. In such cases, the frequency-domain balance equation 24 reduces to

$$\frac{\mathrm{d}}{\mathrm{d}t}E(t) = -2\nu \int_0^\infty k^2 E(k,t)\,\mathrm{d}k = -2\nu \mathcal{E}(t). \tag{27}$$

Using equation 23, we now introduce an energy-weighted average frequency

$$k_E^2(t) := \frac{\displaystyle\int_0^\infty k^2 E(k,t)\,\mathrm{d}k}{\displaystyle\int_0^\infty E(k,t)\,\mathrm{d}k} = \frac{\mathcal{E}(t)}{E(t)}. \tag{28}$$

Then equation 27 may be written as the scalar ODE

$$\frac{\mathrm{d}}{\mathrm{d}t}E(t) = -2\nu k_E^2(t)\, E(t). \tag{29}$$

This linear ODE has the exact solution

$$E(t) = E(0) \exp\Big(-2\nu \int_0^t k_E^2(s)\, \mathrm{d}s\Big),$$
$$\approx E(0) \exp\big(-2\nu k_E^2\, t\big), \tag{30}$$

if the average frequency $k_E(t)$ can be regarded as approximately constant (for instance, in a statistically steady regime).

The equation 30 highlights the joint role of energy and enstrophy in controlling the evolution of the flow: the decay rate of $E(t)$ is governed by the cumulative integral of $k_E^2(t)$, i.e. by how strongly the energy is biased toward small or large scales. When the spectrum is concentrated at large frequency (large $k_E(t)$), kinetic energy decays rapidly; when most energy sits at large scales (small $k_E(t)$), the decay is much slower. These results consist with the implication of equation 6 in the main context.

### C.4 GRADIENT-PENALIZED LOSS AS AN ENERGY-ENSTROPHY TRADE-OFF

We finally explain how our gradient-penalized loss helps mitigate amplitude shrinkage. We consider the continuous version of gradient-penalized loss function 7 on the domain $\Omega$:

$$\mathcal{L}_{\mathrm{gp}}(\tilde{\mathbf{u}}, \mathbf{u}) = \int_\Omega \|\tilde{\mathbf{u}}(x) - \mathbf{u}(x)\|^2\, \mathrm{d}x + \lambda \Big| \|\nabla \tilde{\mathbf{u}}\|_{L^2}^2 - \|\nabla \mathbf{u}\|_{L^2}^2 \Big| \tag{31}$$

where $\tilde{\mathbf{u}}$ is the prediction of the ground-truth $\mathbf{u}$, and $\lambda > 0$ is a tunable hyperparameter. For the vector fields $\nabla \tilde{\mathbf{u}}$ and $\nabla \mathbf{u}$, we have

$$\|\nabla \tilde{\mathbf{u}}\|_{L^2}^2 = \int_\Omega \|\nabla \tilde{\mathbf{u}}(x)\|^2\, \mathrm{d}x, \qquad \|\nabla \mathbf{u}\|_{L^2}^2 = \int_\Omega \|\nabla \mathbf{u}(x)\|^2\, \mathrm{d}x.$$

**Enstrophy matching.** By Theorem 2, we know for a sufficiently regular and divergence-free 2D velocity field $\mathbf{v}$ on $\Omega$,

$$\int_\Omega \|\nabla \mathbf{v}(x)\|^2\, \mathrm{d}x = \int_\Omega |\omega_{\mathbf{v}}(x)|^2\, \mathrm{d}x = \mathcal{E}(\mathbf{v})$$

where $\omega_{\mathbf{v}} = \nabla \times \mathbf{v}$ denotes the vorticity and $\mathcal{E}(\cdot)$ denotes the enstrophy. Consequently, the second term in equation 31 can be interpreted as a *global enstrophy matching* constraint:

$$\Big| \|\nabla \tilde{\mathbf{u}}\|_{L^2}^2 - \|\nabla \mathbf{u}\|_{L^2}^2 \Big| = \big| \mathcal{E}(\tilde{\mathbf{u}}) - \mathcal{E}(\mathbf{u}) \big|,$$

The loss $\mathcal{L}_{\mathrm{gp}}$ therefore penalizes predictions whose overall vorticity and shear intensity is significantly weaker or stronger than that of the reference flow.

**MSE as average energy of the error field.** Let $\mathbf{e}(x) := \tilde{\mathbf{u}}(x) - \mathbf{u}(x)$ denote the error field. Then the MSE term in equation 31 can be written as

$$\mathrm{MSE}(\tilde{\mathbf{u}}, \mathbf{u}) = \int_\Omega \|\mathbf{e}(x)\|^2\, \mathrm{d}x = \|\mathbf{e}\|_{L^2(\Omega)}^2.$$

This quantity is naturally interpreted as the *average kinetic energy* of the error field.

Therefore, the original gradient-penalized loss function can be interpreted from energy perspective as follows:

$$\mathcal{L}_{\mathrm{gp}}(\tilde{\mathbf{u}}, \mathbf{u}) = \underbrace{\int_\Omega \|\tilde{\mathbf{u}}(x) - \mathbf{u}(x)\|^2\, \mathrm{d}x}_{\text{energy matching (MSE)}} + \lambda \underbrace{\Big\lfloor \|\nabla \tilde{\mathbf{u}}\|_{L^2}^2 - \|\nabla \mathbf{u}\|_{L^2}^2 \Big\rfloor}_{\text{enstrophy matching}}, \tag{32}$$

where the "energy matching" means reducing the energy of the error field.

The MSE measures how large the velocity error is in an $L^2$ sense, without distinguishing across spatial scales. In contrast, the enstrophy weights regions with strong gradients and vorticity much more heavily, and is therefore especially sensitive to shear layers, fronts, and small-scale vortices.

**Why enstrophy matching mitigates amplitude shrinkage.** We first conclude the two perspectives of the energy-enstrophy connection analyzed before.

1. By energy balance of the 2D incompressible NS equations22,

$$\frac{1}{2}\frac{\mathrm{d}}{\mathrm{d}t}\|\mathbf{u}(t)\|_{L^2}^2 + \nu\|\nabla\mathbf{u}(t)\|_{L^2}^2 = \langle\mathbf{F}(t), \mathbf{u}(t)\rangle,$$

   we know that the enstrophy governs the viscous dissipation of kinetic energy.

2. By the spectral interpretation of energy and enstrophy 23,

$$E(t) = \frac{1}{2}\int_0^\infty E(k,t)\,\mathrm{d}k, \qquad \mathcal{E}(t) = \int_0^\infty k^2 E(k,t)\,\mathrm{d}k,$$

   we know that the enstrophy is a biased average of spectral energies leaning towards high-frequency spectra.

The above two perspectives jointly provide the properties of enstrophy. A high enstrophy corresponds to strong small-scale activity and rapid energy decay, while low enstrophy corresponds to smoother flows and weaker dissipation. Moreover, enstrophy can be regarded as a measure of "how strongly the flow is twisted and sheared". Since enstrophy weights regions with strong gradients and vorticity much more heavily, it is especially sensitive to shear layers, fronts, and small-scale vortices–the common representation of extreme winds.

In our extreme wind prediction setting, *amplitude shrinkage* manifests precisely in these high-gradient regions: the model tends to underestimate peak wind speeds and over-smooth sharp structures. If training uses only the MSE term, such behavior can be relatively cheap in terms of the objective: the extreme regions typically occupy a small fraction of the domain $\Omega$, so the error introduced by flattening sharp peaks may not dominate the global $L^2$ norm $\|\mathbf{e}\|_{L^2}^2$. As a result, a model that systematically reduces gradients and peak amplitudes can still obtain a low MSE, leading to pronounced amplitude shrinkage in practice.

The enstrophy matching term in equation 13 directly counteracts this tendency. Since the global enstrophy $\|\nabla\mathbf{u}\|_{L^2}^2$ is concentrated in regions where $\|\nabla\mathbf{u}(x)\|$ is large, any systematic smoothing of extreme events and fronts immediately drives $\|\nabla\tilde{\mathbf{u}}\|_{L^2}^2$ significantly below $\|\nabla\mathbf{u}\|_{L^2}^2$. This leads to a large penalty in the regularization term.

Therefore, by using the gradient-penalized loss, the model is encouraged both to:

- maintain accurate average energy of the velocity field via the MSE term, and
- restore sufficient gradient and vorticity strength in high-impact regions so that the total enstrophy $\|\nabla\tilde{\mathbf{u}}\|_{L^2}^2$ remains comparable to $\|\nabla\mathbf{u}\|_{L^2}^2$.

In other words, the loss $\mathcal{L}_{\mathrm{gp}}$ enforces a trade-off between matching the *energy* of the flow and matching its *enstrophy*. This trade-off explicitly discourages solutions that achieve small $L^2$ error by globally smoothing sharp structures and suppressing extremes. Instead, it biases training toward predictions that preserve the correct overall intensity of shear and vorticity, thereby mitigating amplitude shrinkage and yielding more faithful reconstructions of extreme wind events.

## D   CONVERGENCE ANALYSIS OF THE GRADIENT-PENALIZED LOSS MODEL

In this section, we provide a convergence analysis sketch for the simplified model with the gradient-penalized loss function shown in equation 7. The main goal is to explain why excessively large values of $\lambda$ over-amplify high-frequency modes and can destabilize discrete gradient-based optimization, as shown in figure 2.

Recall the gradient-penalized loss

$$\mathcal{L}_{\mathrm{gp}}(\tilde{\mathbf{u}}, \mathbf{u}) = \mathrm{MSE}(\tilde{\mathbf{u}}, \mathbf{u}) + \lambda\left|\|\nabla\tilde{\mathbf{u}}\|^2 - \|\nabla\mathbf{u}\|^2\right|, \qquad\qquad (6\text{ revisited})$$

where $\tilde{\mathbf{u}}$ is the predicted wind field at a given lead time and $\mathbf{u}$ is the ground truth. As discussed in Sec 3.1, in the typical amplitude-shrinkage regime we have $\|\nabla\tilde{\mathbf{u}}\|^2 < \|\nabla\mathbf{u}\|^2$; in this case,

minimizing the original loss function is approximately equivalent to minimizing the following:

$$\mathcal{L}_{\text{gp}}(\tilde{\mathbf{u}}, \mathbf{u}) \;=\; \text{MSE}(\tilde{\mathbf{u}}, \mathbf{u}) \;-\; \lambda \left\| \nabla \tilde{\mathbf{u}} \right\|^2, \tag{8 revisited}$$

which means that, up to an additive constant independent of $\tilde{u}$, the gradient term contributes a negative Sobolev-type penalty on $\|\nabla \tilde{\mathbf{u}}\|^2$. We also recall that

$$\left\| \nabla \tilde{\mathbf{u}} \right\|^2 \;\propto\; \sum_k \|k\|^2 \left\| \hat{\tilde{\mathbf{u}}}(k) \right\|_2^2,$$

where $\hat{\tilde{\mathbf{u}}}(k)$ denotes the discrete Fourier transform of $\tilde{u}$ and $k = (k_x, k_y)$ is the (double) frequency index.

According to Appendix B, the MSE can be equivalently written (up to a positive constant factor) in the frequency domain:

$$\text{MSE}(\tilde{\mathbf{u}}, \mathbf{u}) \;\propto\; \sum_k \left\| \hat{\mathbf{u}}(k) - \hat{\tilde{\mathbf{u}}}(k) \right\|_2^2. \tag{33}$$

Therefore, by absorbing positive constants into $\lambda$ for notational simplicity, we obtain the following normalized local surrogate for $\mathcal{L}_{\text{gp}}$:

$$\tilde{\mathcal{L}}_{\text{gp}}(\tilde{\mathbf{u}}, \mathbf{u}) \;\equiv\; \sum_k \left[ \left\| \hat{\mathbf{u}}(k) - \hat{\tilde{\mathbf{u}}}(k) \right\|_2^2 - \lambda \|k\|^2 \left\| \hat{\tilde{\mathbf{u}}}(k) \right\|_2^2 \right]. \tag{34}$$

Thus the loss is decomposed into a sum of independent vector-valued quadratics over frequency spectra.

For the clarity of notations, we now denote $\mathbf{p}_k := \hat{\tilde{\mathbf{u}}}(k)$ and $\mathbf{q}_k := \hat{\mathbf{u}}(k) \in \mathbb{R}^d$ for every frequency spectrum $k$. Then the contribution of each spectrum $k$ to $\mathcal{L}_{gp}$ will be

$$\ell_k(\mathbf{p}_k) \;=\; \left\| \mathbf{p}_k - \mathbf{q}_k \right\|_2^2 \;-\; \lambda \|k\|^2 \left\| \mathbf{p}_k \right\|_2^2. \tag{35}$$

## D.1 Vector Hessian eigenvalues analysis

We now compute the gradient and Hessian of $\ell_k$ with respect to $\mathbf{p}_k$. Expanding the squared norms in equation 35 gives

$$\ell_k(\mathbf{p}_k) = \mathbf{p}_k^\top \mathbf{p}_k - 2 \mathbf{q}_k^\top \mathbf{p}_k + \mathbf{q}_k^\top \mathbf{q}_k - \lambda \|k\|^2 \mathbf{p}_k^\top \mathbf{p}_k.$$

Differentiating with respect to $\mathbf{p}_k$ yields

$$\nabla_{\mathbf{p}_k} \ell_k = 2\mathbf{p}_k - 2\mathbf{q}_k - 2\lambda \|k\|^2 \mathbf{p}_k = 2 \big[ (1 - \lambda \|k\|^2) \mathbf{p}_k - \mathbf{q}_k \big].$$

Differentiating once again, we get the Hessian of $\ell_k$:

$$\nabla_{\mathbf{p}_k}^2 \ell_k = 2(1 - \lambda \|k\|^2) I_d, \tag{36}$$

where $I_d$ is the identity matrix. Thus, all $d$ eigenvalues of the Hessian block for mode $k$ are equal to

$$h_k(\lambda) = 2(1 - \lambda \|k\|^2) \begin{cases} > 0, & \text{if } \lambda \|k\|^2 < 1, \\ = 0, & \text{if } \lambda \|k\|^2 = 1, \\ < 0, & \text{if } \lambda \|k\|^2 > 1. \end{cases} \tag{37}$$

The above results imply the theoretical convexity of the simplified gradient-penalized loss function. For low frequencies (small $\|k\|$) and moderate $\lambda$, we have $h_k(\lambda) > 0$ and the loss remains locally convex in those directions. However, for sufficiently large $\lambda$, modes with $\lambda \|k\|^2 > 1$ exhibit negative curvature and the local quadratic approximation becomes a saddle (or even a local maximum) along those high-frequency directions. Practically, the extremely high-frequency components of the data are negligible. Therefore, we may assume that the frequency support is bounded by some constant $K_{\max}$, i.e. $\|k\| \le K_{\max}$. In these cases, a sufficient condition for (block-diagonal) positive definiteness of the Hessian of $\tilde{\mathcal{L}}_{\text{gp}}$ is

$$1 - \lambda K_{\max}^2 > 0 \qquad \Longleftrightarrow \qquad \lambda < \frac{1}{K_{\max}^2}. \tag{38}$$

## D.2 GRADIENT DESCENT DYNAMICS AND INSTABILITY FOR LARGE $\lambda$

We now analyze vanilla gradient descent dynamics on the per-spectrum objective function $\ell_k(\mathbf{p}_k)$ with step size $\eta > 0$. The update rule is

$$\mathbf{p}_k^{(t+1)} = \mathbf{p}_k^{(t)} - \eta \nabla_{\mathbf{p}_k} \ell_k(\mathbf{p}_k^{(t)}). \tag{39}$$

Suppose there exists a stationary point $\mathbf{p}_k^*$ for $\ell_k$. Defining the error $\mathbf{e}_k^{(t)} := \mathbf{p}_k^{(t)} - \mathbf{p}_k^*$, we have the following lemma:

**Lemma 1** (Spectrum stability condition). *Under the update rule 39, the error $\mathbf{e}_k^{(t)}$ converges to zero only if*

$$0 < \eta(1 - \lambda\|k\|^2) < 1, \tag{40}$$

*or equivalently,*

$$\begin{cases} 1 - \lambda\|k\|^2 > 0, \\ 0 < \eta < \dfrac{1}{1 - \lambda\|k\|^2}. \end{cases} \tag{41}$$

*Proof.* Plugging the gradient of the spectral loss function $\nabla_{\mathbf{p}_k} \ell_k$ into 39, we have

$$\mathbf{p}_k^{(t+1)} = \mathbf{p}_k^{(t)} - \eta \nabla_{\mathbf{p}_k} \ell_k(\mathbf{p}_k^{(t)}) = \mathbf{p}_k^{(t)} - 2\eta\big[(1 - \lambda\|k\|^2)\mathbf{p}_k^{(t)} - \mathbf{q}_k\big]. \tag{42}$$

Therefore, a stationary point $\mathbf{p}_k^*$ should satisfy

$$(1 - \lambda\|k\|^2)\mathbf{p}_k^* - \mathbf{q}_k = 0 \qquad \implies \qquad \mathbf{p}_k^* = \frac{1}{1 - \lambda\|k\|^2}\,\mathbf{q}_k,$$

provided $1 - \lambda\|k\|^2 \neq 0$. Therefore, the recurrence formula of $\mathbf{e}_k^{(t)}$ is

$$\mathbf{e}_k^{(t+1)} = \Big[I_d - 2\eta(1 - \lambda\|k\|^2)I_d\Big]\mathbf{e}_k^{(t)} = \rho_k(\lambda, \eta)\,\mathbf{e}_k^{(t)}, \tag{43}$$

with the linear recurrence scaling factor

$$\rho_k(\lambda, \eta) := 1 - \eta\, h_k(\lambda) = 1 - 2\eta(1 - \lambda\|k\|^2).$$

Since the update matrix is a scalar multiple of $I_d$, each component of $\mathbf{e}_k^{(t)}$ follows the same one-dimensional recurrence. The recurrence formula 43 converges only if $|\rho| < 1$, which is equivalent to

$$-1 < 1 - 2\eta(1 - \lambda\|k\|^2) < 1 \iff 0 < 2\eta(1 - \lambda\|k\|^2) < 2,$$

which finishes the proof. $\qquad\square$

**Interpretation.** Lemma 1 reveals two key effects of $\lambda$ in the vector-valued Fourier setting:

(i) **Theoretical upper-bound of $\lambda$.** For any spectrum $k$ satisfying $\lambda\|k\|^2 \geq 1$, we have $1 - \lambda\|k\|^2 \leq 0$ and no choice of $\eta > 0$ can satisfy $0 < \eta(1 - \lambda\|k\|^2) < 1$. In this case the per-mode quadratic is flat or concave and gradient descent cannot converge to a local minimum along that high-frequency direction. The above analysis suggests that there exists a theoretical upper bound $\lambda_{\max} \leq 1/K_{\max}^2$, where $K_{\max}$ denotes a high-frequency cutoff chosen so that the majority of the spectral energy is concentrated in spectra with $\|k\| \leq K_{\max}$. If $\lambda$ is chosen larger than the upper-bound, there is no guarantee that the training will converge. A necessary condition for $\lambda$ and the learning rate should be

$$\lambda \leq \lambda_{\max} < \frac{1}{K_{\max}^2} \quad \text{and} \quad 0 < \eta < \frac{1}{1 - \lambda K_{\max}^2} \tag{44}$$

As a result, the performance as a function of $\lambda$ naturally exhibits a U-shaped curve, as illustrated by figure 2: small to moderate $\lambda$ improves the fit by correcting amplitude shrinkage, while overly large $\lambda$ will degrade or even destabilize the optimization and affect final accuracy.

(ii) **High-frequency spectra suffer more.** Given that $\lambda$ is fixed, the effective spectral-wise curvature $h_k(\lambda) = 2(1 - \lambda\|k\|^2)$ is more likely to become negative for high-frequency spectra. Therefore, if $\lambda$ is not chosen properly, it's possible to witness the instability of training in regions with large energy of higher-frequency spectra, such as turbulent regions.

**Remark.** This convergence sketch is based on a simplified model of the gradient-penalized loss rather than the full nonlinear network and training pipeline. Although real-world optimization dynamics are more complex, the above analysis reveals the key structural effect of our method, and provides the intuitive insights of the U-shape performance curve with respect to $\lambda$, in line with the empirical experiments.

# E  DATASET AND EVALUATION METRICS

## E.1  ERA5 DATASET

We mainly used the ERA5 dataset for our model training and testing processes. The ERA5 dataset(Hersbach et al., 2020), developed by the European Centre for Medium-Range Weather Forecasts (ECMWF), is a fifth-generation reanalysis of the climate and weather covering data from 1940 to the present. Although the dataset contains detailed reanalysis data globally, it provides flexibility to select and obtain data in rectangular spatial region in different scales and locations. Therefore, the dataset is suitable for studying our work on regional weather prediction. This dataset is created through data assimilation, which combines model data with observations from various sources worldwide, resulting in a globally consistent and comprehensive dataset. ERA5 provides hourly estimates for a wide range of atmospheric, ocean-wave, and land-surface variables, including uncertainty estimates using a 10-member ensemble at three-hour intervals. The data is available on a regular latitude-longitude grid, with a horizontal resolution of $0.25° \times 0.25°$ for atmospheric reanalysis. The temporal resolution of ERA5 is hourly, and the data is accessible in GRIB format, providing high-resolution information for many climate and weather applications.

In this study, we focus on specific variables from the ERA5 dataset relevant to wind speed prediction, namely the 10-meter wind components and surface pressure. The 10-meter u-component of wind represents the eastward component of horizontal wind speed at 10 meters above ground level, while the v-component represents the northward component at the same height. These components are measured in meters per second (m/s) and can be combined to calculate the speed and direction of the horizontal wind. Surface pressure, given in Pascals (Pa), is the atmospheric pressure at the Earth's surface, which reflects the weight of the air column above a specific point. These parameters together provide essential information for modeling and predicting wind dynamics in the atmosphere.

## E.2  DETAILS OF EVALUATION METRICS

**Root Mean Squared Error (RMSE)**  The RMSE quantifies the overall accuracy of the predicted wind velocity field by measuring the difference between the predicted and ground truth values. It is defined as:

$$\text{RMSE} = \sqrt{\frac{1}{N} \sum_{i=1}^{N} \|\hat{\mathbf{u}}_i - \mathbf{u}_i\|^2},$$

where:

- $\hat{\mathbf{u}}_i$ and $\mathbf{u}_i$ are the predicted and ground truth wind velocity vectors at the $i$-th grid point,
- $N$ is the total number of grid points.

**Extreme Region Error (Ex-RMSE)**  The Extreme Region Error (Ex-RMSE) focuses on the model's accuracy in predicting extreme weather regions, characterized by high wind velocities. It assigns larger weights to regions with extreme wind values to emphasize their importance. Mathematically, it is defined as:

$$\text{ExtremeErr} = \sqrt{\frac{\sum_{i=1}^{N} w_i \cdot \|\hat{\mathbf{u}}_i - \mathbf{u}_i\|^2}{\sum_{i=1}^{N} w_i}},$$

where:

- $w_i$ is the weight assigned to the $i$-th grid point, with higher values for extreme wind velocity regions,

- $\hat{\mathbf{u}}_i$ and $\mathbf{u}_i$ are the predicted and ground truth wind velocity vectors at the $i$-th grid point.

These metrics collectively assess the model's accuracy, adherence to physical principles, and capability to predict extreme weather conditions effectively.

# F    ADDITIONAL EXPERIMENTS

## F.1    IMPACT OF FREQUENCY ON TEMPORAL DATA

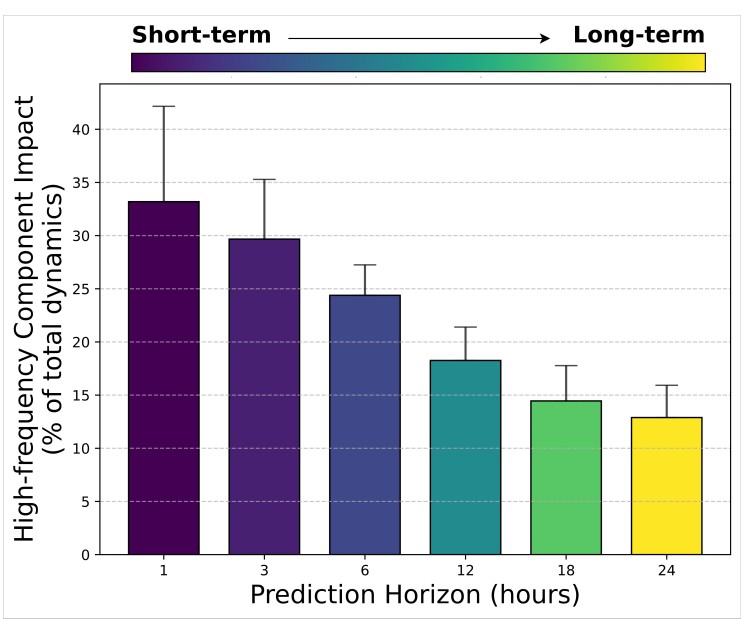

Figure 4: Impact of high-frequency components on wind field dynamics decreases as prediction horizon extends from 1 to 24 hours.

We conducted experiments to investigate how high- and low-frequency components of wind data contribute to future dynamics across different temporal scales. Using Fourier filtering techniques (detailed in Section 4), we decomposed the two-dimensional wind velocity field time series into their respective frequency components.

Our correlation analysis examined the relationship between these decomposed components and actual future wind patterns across prediction horizons ranging from 1 to 24 hours. The results, illustrated in Figure 4, reveal a clear temporal dependency pattern: For longer prediction horizons (approaching 24 hours), low-frequency components demonstrate dominant predictive power in wind speed pattern evolution. Conversely, at shorter intervals (approaching 1 hour), high-frequency components become increasingly significant in determining wind pattern changes.

This observation can be theoretically explained as follows. Suppose $\mathbf{u}(x, y)$ is the wind velocity field. Then the Fourier transform of the spatial gradient of wind velocity $\nabla \mathbf{u}$ is:

$$\nabla \hat{\mathbf{u}}(\mathbf{k}) = i\mathbf{k} \iint u(x,y) e^{-i2\pi(k_x x + k_y y)} \, dx \, dy,$$

where $k = (k_x, k_y)$ represents frequency domain coordinates and $\hat{\cdot}$ denotes the Fourier transform. This relationship demonstrates that higher frequencies (larger $|k|$) correspond to larger spatial gradients ($\|\nabla u\|$). Consequently, high-frequency components capture small-scale features characterized by sharp gradients and abrupt changes in the wind velocity field—characteristics typically associated with turbulence and extreme weather events.

## F.2    DIFFERENT FREQUENCY MASKING LEVEL

In this subsection, we investigate how different frequency masking levels affect the model's wind velocity prediction performance using the Fourier Frequency Filter. The results show that excessively high or low masking thresholds degrade accuracy, while optimal performance is achieved at intermediate levels, where a balance between high- and low-frequency information is maintained.

To explore this, we varied the threshold for dividing high- and low-frequency components and analyzed its effect on wind speed prediction accuracy. Experiments were conducted using frequency masking levels of 0.1, 0.3, 0.5, 0.7, and 0.9, which represent the proportion of the highest frequencies included in the high-frequency data. These experiments were performed on wind velocity field data from four distinct regions, with the results summarized in Figure 5.

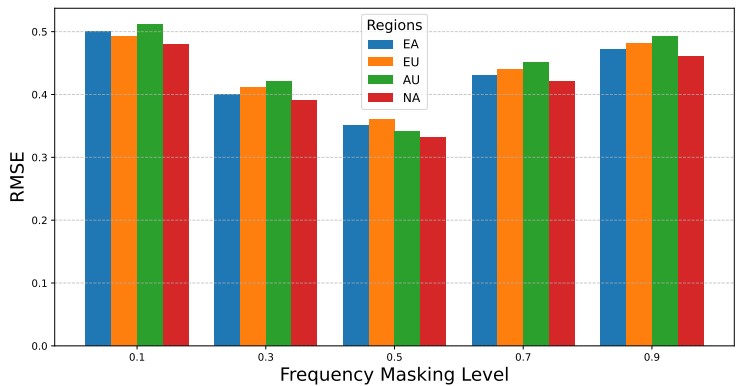

Figure 5: Impact of frequency masking levels on RMSE across different regions.

The results demonstrate that both excessively high and excessively low-frequency masking thresholds negatively impact the model's prediction accuracy. When the masking level is too high, critical low-frequency information is excluded, leading to incomplete data representation. Conversely, when the masking level is too low, significant high-frequency details are overlooked, impairing the model's ability to capture rapid variations in wind speed. Optimal prediction performance is achieved when the frequency masking level lies between 0.3 and 0.7, as this range effectively balances the inclusion of high- and low-frequency information, enabling the model to better capture both large-scale and small-scale dynamics.

## F.3    CROSS REGIONAL EXPERIMENTS

To further evaluate the robustness and generalizability of our model, we conducted additional experiments using datasets from different geographical regions. The experimental setup, including the fundamental parameters and evaluation metrics, remained consistent with those described in the main text to ensure comparability.

Among the various results obtained, we present the most representative findings in Table 3 and Figure 6. In these experiments, EA, NA, and AU correspond to datasets from East Asia, North America, and Australia, respectively. The results consistently demonstrate that our model outperforms the baseline and comparative methods in predicting extreme wind speeds across all tested regions. Specifically, our model achieves the lowest RMSE while maintaining high accuracy in capturing the most significant variations in wind speed. Moreover, the improvements are particularly evident in regions with frequent extreme weather events, further validating the effectiveness of our approach in handling complex wind dynamics.

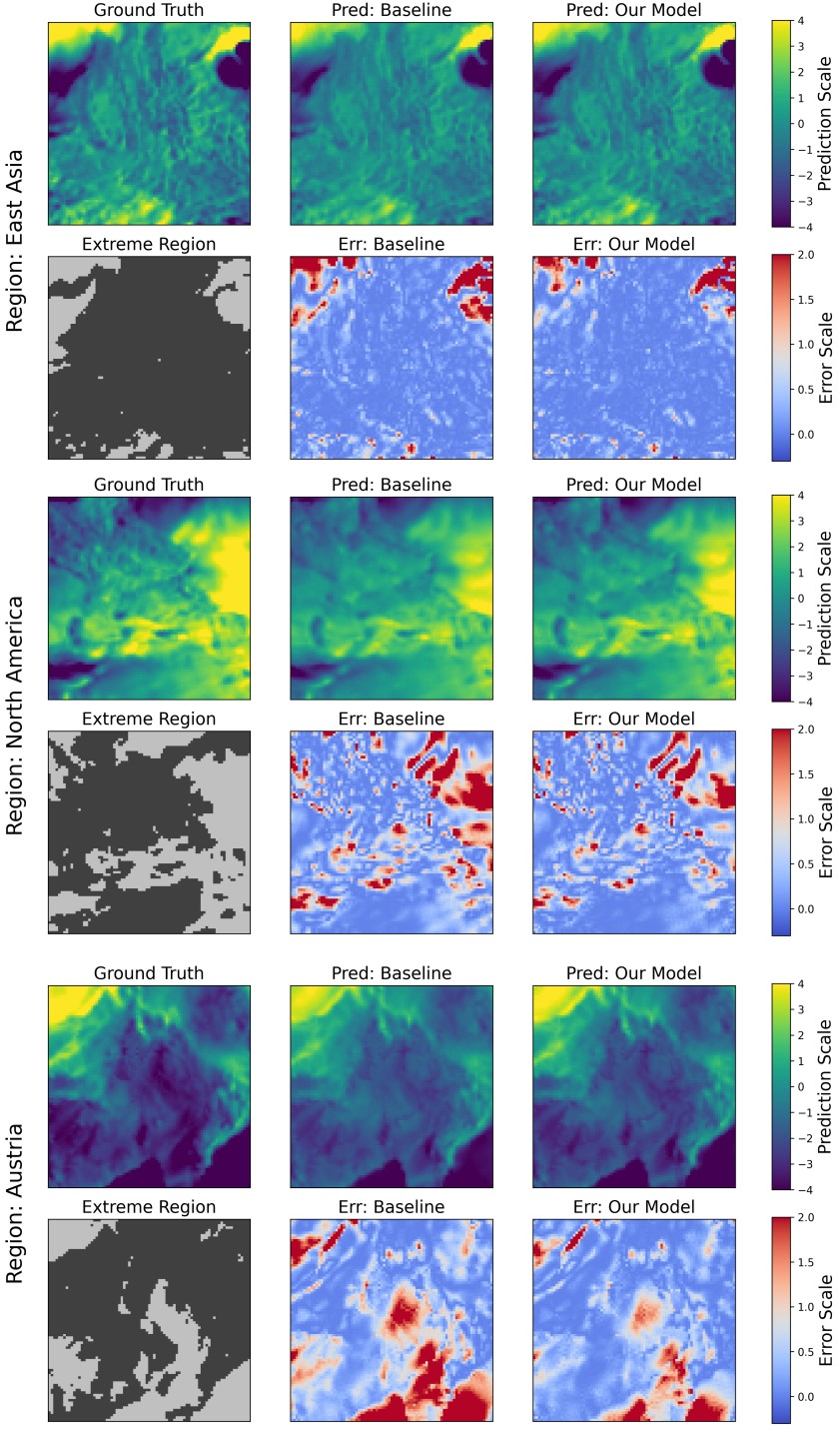

Figure 6: Results of cross-regional experiments.

Table 3: Comparative error results across regions and models.

| Region | Model | Lead time: 1h | | Lead time: 3h | | Lead time: 5h | |
|---|---|---|---|---|---|---|---|
| | | RMSE | Ex-RMSE | RMSE | Ex-RMSE | RMSE | Ex-RMSE |
| EA | CNN | 0.9559 | 0.6011 | 3.3106 | 2.2872 | 5.1008 | 3.7931 |
| | ConvLSTM | 0.8356 | 0.4217 | 2.4391 | 1.5702 | 3.0405 | 2.5460 |
| | PINN | 0.8156 | 0.3983 | 2.0649 | 1.4828 | 2.8327 | 2.2321 |
| | **Ours** | **0.6947** | **0.3236** | **1.8657** | **1.3334** | **2.5840** | **1.9037** |
| NA | CNN | 0.8309 | 0.5691 | 1.9977 | 1.5617 | 3.6052 | 2.4897 |
| | ConvLSTM | 0.6569 | 0.4237 | 1.2097 | 1.0331 | 1.7852 | 1.5028 |
| | PINN | 0.6630 | 0.4863 | 1.3394 | 1.1201 | 1.8592 | 1.6391 |
| | **Ours** | **0.6065** | **0.3872** | **1.1935** | **0.9305** | **1.6862** | **1.2477** |
| AU | CNN | 1.1043 | 0.6830 | 2.3372 | 1.4968 | 3.0311 | 2.1449 |
| | ConvLSTM | 0.8584 | 0.3292 | 2.0674 | 1.0791 | 2.4728 | 1.7153 |
| | PINN | 0.9342 | 0.4409 | 2.2171 | 1.0831 | 2.3327 | 1.5570 |
| | **Ours** | **0.7411** | **0.3244** | **1.8853** | **0.8901** | **2.0521** | **1.1371** |

