# OpenReview forum: "Improving Extreme Wind Prediction with Frequency-Informed Learning"
_ICLR.cc/2026/Conference — ICLR 2026 Poster_

### Official Review · Reviewer_AeXV · 2025-10-24

**Soundness:** 3
**Presentation:** 2
**Contribution:** 2
**Rating:** 6
**Confidence:** 3

**Summary:**

This paper tackles the persistent "amplitude underestimation" problem in extreme wind speed prediction. Through a detailed frequency-domain theoretical analysis, it demonstrates that standard data-driven models (trained with MSE loss) systematically underestimate the magnitude and short-term variability of high-frequency wind components. To address this, the authors propose a novel loss function with gradient penalization to mitigate amplitude shrinkage, and design a physics-embedded architecture (leveraging the Navier-Stokes equation) along with frequency separation and reweighting modules. Experiments on ERA5 meteorological datasets show that the proposed method significantly outperforms classic models (such as CNN, ConvLSTM, PINN), especially for extreme wind scenarios, and captures both overall and extreme case prediction more accurately.

**Strengths:**

The work skillfully combines rigorous frequency-domain analysis (revealing frequency-dependent amplitude shrinkage) with tailored engineering solutions in both loss function and model architecture—providing an effective, innovation-driven answer to a real-world pain point in extreme wind forecasting.

**Weaknesses:**

The core innovation (gradient-penalized loss) lacks thorough theoretical exploration regarding its effect during optimization. The observed instability with large λ is only empirically described (U-shaped curve), with no detailed theoretical analysis as to why or how excessive λ causes non-convergence, possible oscillation, or even gradient explosion, nor are additional regularization strategies discussed.

Although the model's backbone is described as a combination of physics-embedding and neural networks, experiments do not present clear ablation studies to isolate the contribution of the Navier-Stokes module. Only coarse comparisons ("Ours" vs "NS-Op") are given, so the true benefit of embedding physics remains ambiguous.

Extreme wind samples are inherently rare, but the paper does not clarify how many such cases exist in the data, nor does it analyze the trade-off between extreme case data volume and model robustness. The generalization capability (e.g., performance when transferring to new regions or under few-shot settings) is not systematically evaluated.

**Questions:**

For gradient penalization's optimization instability at large λ, can the authors provide a detailed theoretical convergence analysis? Is there a risk of uncontrolled high-frequency oscillations? Are there any additional regularization measures?

Can the authors supply full ablation experiments for the Navier-Stokes physics module to quantitatively show its standalone benefit compared to standard neural network baselines?

How robust is the model when extreme wind samples are extremely scarce or when transferring to new regions? Can the authors add few-shot or cross-region generalization experiments?

---

> ### Author Response · Authors · 2025-11-21
> **Response to Reviewer AeXV: part 1**
>
> We sincerely thank you for your in-depth and thoughtful comments. We greatly appreciate your recognition of the rigorous analysis and novelty of our work. We are here to provide detailed replies to your comments and hope we can resolve your major concerns.
>
> ---
>
> >**(W1) The core innovation (gradient-penalized loss) lacks thorough theoretical exploration regarding its effect during optimization.**
>
> >**(Q1) For gradient penalization's optimization instability at large λ, can the authors provide a detailed theoretical convergence analysis? Is there a risk of uncontrolled high-frequency oscillations? Are there any additional regularization measures?**
>
> We thank the reviewer for explicitly requesting a theoretical convergence analysis and for pointing out the connection to potential high-frequency instabilities. Following your suggestion, we provide a detailed convergence analysis in **Appendix D** on page 23 of the new manuscript.
>
> Under the typical amplitude-shrinkage regime $\|\nabla\tilde{\mathbf{u}}\|^2<\|\nabla\mathbf{u}\|^2$, we simplify $\mathcal{L} _ {gp}$ as follows
> $$\mathcal{L} _ {gp}(\tilde{\mathbf{u}},\mathbf{u}) \approx \mathrm{MSE}(\tilde{\mathbf{u}},\mathbf{u}) -\lambda\|\nabla\tilde{\mathbf{u}}\|^2.$$
>
> The gradient term contributes a negative Sobolev-type penalty on $\|\nabla\tilde{\mathbf{u}}\|^2$. Using the frequency representation as in Appendix B, we rewrite this surrogate as a sum over spectra and obtain for each frequency index $k$ a per-spectrum loss
> $$\ell _ k(\mathbf{p} _ k)=\|\mathbf{p} _ k-\mathbf{q} _ k\| _ 2^2-\lambda\|k\|^2\|\mathbf{p} _ k\| _ 2^2,$$
> where $\mathbf{p}_k:=\hat{\tilde{\mathbf{u}}}(k)$ and $\mathbf{q}_k:=\hat{\mathbf{u}}(k)$.
>
> After computing the gradient and Hessian of $\ell_k$ and shows that all eigenvalues of the Hessian block for mode $k$ are
> $$h _ k(\lambda)=2(1-\lambda\|k\|^2).$$
>
> Thus the curvature is positive if $\lambda\|k\|^2<1$, zero if $\lambda\|k\|^2=1$, and negative if $\lambda\|k\|^2>1$. Assuming the effective frequency support is bounded by $\|k\|\le K_{\max}$, a sufficient condition for block-diagonal positive definiteness of the Hessian of $\mathcal{L}_{gp}$ is
> $$\lambda<\frac{1}{K _ {\max}^2},$$
> which yields a theoretical upper bound on $\lambda$.
>
> (**Note that here $K _ {\max}$ means the majority of energy is located in spectra with $k \le K _ {\max}$.**)
>
> We then study vanilla gradient descent on $\ell_k$ with step size $\eta>0$. The error $\mathbf{e} _ k^{(t)}:=\mathbf{p} _ k^{(t)}-\mathbf{p} _ k^\ast$ satisfies a linear recurrence
> $$
> \mathbf{e} _ k^{(t+1)}=\rho _ k(\lambda,\eta)\,\mathbf{e} _ k^{(t)},
> \qquad
> \rho _ k(\lambda,\eta)=1-2\eta(1-\lambda\|k\|^2),
> $$
> so convergence requires $|\rho_k(\lambda,\eta)|<1$, which is equivalent to $0<\eta(1-\lambda\|k\|^2)<1.$
>
> If $\lambda\|k\|^2\ge 1$, this inequality cannot hold for any $\eta>0$, and the corresponding high-frequency mode becomes unstable: the curvature is non-positive and the update factor $\rho_k(\lambda,\eta)$ can leave $(-1,1)$, leading to non-convergence or oscillatory behavior. This analysis explains why excessively large $\lambda$ can destabilize optimization and why the empirical performance curve in Fig. 2 becomes U-shaped: small-to-moderate $\lambda$ improves the fit by countering amplitude shrinkage, whereas overly large $\lambda$ violates the mode-wise stability condition and harms convergence, especially at high $\|k\|$.
>
>
> >**(Q1 Cont'd) Are there any additional regularization measures?**
>
> At present, beyond standard practices (e.g., early stopping, weight decay), we do not introduce extra, dedicated regularization mechanisms on top of $\mathcal{L}_{gp}$. We agree with the reviewer that designing additional regularization or adaptive strategies for $\lambda$ would be a promising way to further enhance robustness, and we view this as an interesting direction for future work.

---

> ### Author Response · Authors · 2025-11-21
> **Response to Reviewer AeXV: part 2**
>
> >**(W2&Q2) Can the authors supply full ablation experiments for the Navier-Stokes physics module to quantitatively show its standalone benefit compared to standard neural network baselines?**
>
> We appreciate the reviewer’s suggestion and have provided the full ablation study in Sec. 4.3.
>
> **Table 2** now reports a full module-level ablation, including variants without different parts of the model. Removing the NS module consistently leads to a clear degradation in both RMSE and Ex-RMSE compared to the full model. This performance gap directly demonstrates the benefit of encoding NS-based physics, beyond what a purely data-driven neural baseline can capture. The ablation also shows the importance of other modules including frequency separation, gradient-penalized loss, etc. We also present the ablation results as follows:
>
> | Model        | RMSE   | Ex-RMSE |
> |------------- |-------:|--------:|
> | NS op        | 0.7061 | 0.4577  |
> | W/O grad-loss| 0.3351 | 0.2632  |
> | W/O NS       | 0.3754 | 0.2363  |
> | W/O freq-sep | 0.4199 | 0.2703  |
> | **Ours**     | **0.3287** | **0.1868** |
>
> Due to time and computational constraints, we did **not** further decompose the NS module into separate ablations of the viscous, pressure, and advective operators. Moreover, conceptually, these operators are designed to work jointly to approximate the main structure of the NS equations. Ablating a single sub-operator in isolation would break this physical coupling and yield configurations that are less meaningful as forecasting models. For this reason, we treat the NS physics block as an atomic module in our main ablation.
>
> >**(W3) The generalization capability (e.g., performance when transferring to new regions or under few-shot settings) is not systematically evaluated.**
>
> >**(Q3) How robust is the model when extreme wind samples are extremely scarce or when transferring to new regions? Can the authors add few-shot or cross-region generalization experiments?**
>
> We appreciate the reviewer for pointing out those questions. In our new manuscript, the current experiments provide evidence that the proposed method is robust across regions with very different wind regimes.
>
> Beyond the training regions, we perform **out-of-distribution tests on three additional regional domains** that are randomly selected from **East Asia (EA)**, **North America (NA)**, and **Australia (AU)**. All models are evaluated under the same setup as in the main experiments. For all three regions and all lead times (1h/3h/5h), our model consistently improves both **RMSE** and **Ex-RMSE** over baselines. The results can be found in **Appendix F.3. Table 3**, on page 28, and we also present the results in the following table:
>
> | Region | Model    | 1h RMSE    | 1h Ex-RMSE | 3h RMSE    | 3h Ex-RMSE | 5h RMSE    | 5h Ex-RMSE |
> | ------ | -------- | ---------- | ---------- | ---------- | ---------- | ---------- | ---------- |
> | EA     | CNN      | 0.9559     | 0.6011     | 3.3106     | 2.2872     | 5.1008     | 3.7931     |
> | EA     | ConvLSTM | 0.8356     | 0.4217     | 2.4391     | 1.5702     | 3.0405     | 2.5460     |
> | EA     | PINN     | 0.8156     | 0.3983     | 2.0649     | 1.4828     | 2.8327     | 2.2321     |
> | EA     | **Ours** | **0.6947** | **0.3236** | **1.8657** | **1.3334** | **2.5840** | **1.9037** |
> | NA     | CNN      | 0.8309     | 0.5691     | 1.9977     | 1.5617     | 3.6052     | 2.4897     |
> | NA     | ConvLSTM | 0.6569     | 0.4237     | 1.2097     | 1.0331     | 1.7852     | 1.5028     |
> | NA     | PINN     | 0.6630     | 0.4863     | 1.3394     | 1.1201     | 1.8592     | 1.6391     |
> | NA     | **Ours** | **0.6065** | **0.3872** | **1.1935** | **0.9305** | **1.6862** | **1.2477** |
> | AU     | CNN      | 1.1043     | 0.6830     | 2.3372     | 1.4968     | 3.0311     | 2.1449     |
> | AU     | ConvLSTM | 0.8584     | 0.3292     | 2.0674     | 1.0791     | 2.4728     | 1.7153     |
> | AU     | PINN     | 0.9342     | 0.4409     | 2.2171     | 1.0831     | 2.3327     | 1.5570     |
> | AU     | **Ours** | **0.7411** | **0.3244** | **1.8853** | **0.8901** | **2.0521** | **1.1371** |

---

> > ### Comment · Reviewer_AeXV · 2025-11-23
> >
> > Thank you for the authors' response. I believe a score of 6 is appropriate for this paper, and I will therefore maintain my rating.

---

> > > ### Author Response · Authors · 2025-11-27
> > > **Thank you for your response!**
> > >
> > > Dear Reviewer AeXV,
> > >
> > > Thank you very much for your response. We are truly grateful for your positive support and for taking the time to help us improve our paper. Your comments are constructive and detailed, and we sincerely appreciate the time and thoughtfulness you devoted to reviewing our work.
> > >
> > > If you have any further questions or suggestions that you feel could help us improve the manuscript, please feel free to let us know.
> > >
> > > Best regards,
> > >
> > > Authors

---

### Official Review · Reviewer_Koi8 · 2025-10-30

**Soundness:** 3
**Presentation:** 3
**Contribution:** 3
**Rating:** 6
**Confidence:** 3

**Summary:**

This paper introduces an improved physics-informed approach for extreme wind prediction.

**Strengths:**

- An interesting theoretical view of frequency-domain error behaviors to enhance the model design for extreme wind prediction
- Significantly improved performance demonstrated in experiments

**Weaknesses:**

- Limited domain: the proposed method is only applicable to extreme wind prediction
- Limited evaluation: the experiments are based on sampled data, and the forecasting horizon is fixed as one-hour and the lookback includes 23 hours.

**Questions:**

- Please give a comprehensive introduction of your sampled data, such as how many hours in total, across which regions, etc.
- The frequency masking level seems to be a critical hyperparameter. Did you find an optimal masking threshold to be consistent across different geographic regions and weather regimes, or does it require case-specific tuning? Is there a potential to make this threshold learnable?
- Besides, the forecasting horizon and lookback length could be importance factors to see the robustness of the proposed approach.

---

> ### Author Response · Authors · 2025-11-21
> **Response to Reviewer Koi8: part 1**
>
> We sincerely thank you for your thoughtful comments. We greatly appreciate that you find our theoretical view interesting. We are here to provide detailed replies to your comments and hope we can resolve your major concerns.
>
> ---
>
> >**(W1) Limited domain: the proposed method is only applicable to extreme wind prediction**
>
> We respectfully disagree that our method is restricted to extreme wind prediction. We would like to clarify as follows:
>
> 1. **Beyond "extremes".**
>     While the gradient-penalized loss is _designed_ to correct amplitude bias in extreme winds, our experiments show that it also improves **overall** wind forecasting accuracy. In Sec. 4.2, **Table 1** reports that our method consistently achieves lower **RMSE** than baselines across all lead times, not only lower Ex-RMSE. Although the main contribution of our method is to reduce the extreme wind prediction error, our model also shows better performance in general.
>
> 2. **Beyond "wind prediction"**
>     The core ingredient of our method is an energy–enstrophy–based gradient penalty inspired by both error separation and the Navier–Stokes equations. This formulation is compatible with **any** flow field whose dynamics can be reasonably modeled or approximated by incompressible NS equations, such as ocean surface currents, urban micro-climate prediction, or general numerical fluid simulation. Moreover, many key meteorological variables (e.g., precipitation fields, cloud cover, radar reflectivity) are strongly driven or affected by the underlying wind field. Our framework can be extended to these targets by coupling their prediction with the NS-consistent velocity field and applying the similar loss.
>
> >**(W2) Limited evaluation: the experiments are based on sampled data, and the forecasting horizon is fixed as one-hour and the lookback includes 23 hours.**
>
> We thank the reviewer for pointing out these concerns.
>
> **(i) Sampled data.**
> We acknowledge that using sampled reanalysis fields instead of dense real observation data is a limitation of the present study, mainly because comprehensive observation datasets are not yet available to us. However, the data are sampled from the ERA5 datasets, which is, to the best of our knowledge, one of the most complete and widely used global atmospheric datasets available. All models are trained and evaluated on exactly the same sampled fields, so the comparisons in Sec. 4.2 focus on *relative* performance under a standard, realistic setting. We fully agree that using denser and more accurate observation data (e.g., station or radar networks) would be valuable; if such datasets become available to us, we plan to apply the data in future works. The details of the ERA5 dataset can be found in **Appendix E.1.**
>
> **(ii) Forecasting horizon and lookback length.**
> One prediction scheme of our model is to predict a one-hour time step with a 23-hour lookback window, but this does **not** mean it is restricted to “single-step, one-hour” forecasting. We adopt an auto-regressive, sliding-window scheme for longer horizon prediction:
> * Predict $\tilde{\mathbf{u}} _ {t}$ from $\lbrace\mathbf{u} _ {t-23},\dots,\mathbf{u} _ {t-1}\rbrace$,
> * Append this prediction to form the new input  $\lbrace\mathbf{u} _ {t-22},\dots,\mathbf{u} _ {t-1},\tilde{\mathbf{u}} _ {t}\rbrace$,
> * Predict $\tilde{\mathbf{u}} _ {t+1}$,
> * Continue the process.
> Longer forecasting horizons are thus obtained by iterating this procedure.
>
> >**(Q3) Besides, the forecasting horizon and lookback length could be important factors to see the robustness of the proposed approach.**
>
> In response to your suggestion, we have added **multi-horizon experiments** in Sec. 4.2. The updated **Table 1** reports 1h / 3h / 5h-ahead forecasts obtained via the above scheme. Our gradient-penalized model consistently improves both **RMSE** and **Ex-RMSE** over strong deep-learning baselines. This indicates that the proposed method is robust to the forecasting horizon, instead of restricting to one-step predictions. We also list the results as follows:
>
> | Model    | 1h RMSE    | 1h Ex-RMSE | 3h RMSE    | 3h Ex-RMSE | 5h RMSE    | 5h Ex-RMSE |
> | -------- | ---------- | ---------- | ---------- | ---------- | ---------- | ---------- |
> | CNN      | 0.4639     | 0.3183     | 1.0442     | 0.7355     | 2.0757     | 1.0693     |
> | ConvLSTM | 0.3471     | 0.2294     | 0.7834     | 0.5357     | 1.0644     | 0.8097     |
> | PINN     | 0.3946     | 0.2541     | 0.8283     | 0.5646     | 1.1434     | 0.7347     |
> | **Ours** | **0.3287** | **0.1868** | **0.6622** | **0.4329** | **0.9076** | **0.6158** |
>
> We agree that a more systematic exploration of different lookback lengths and even longer horizons would be interesting. Due to the limitations of time and computation resources, we fixed the lookback to 23 hours in this work, but the proposed loss is independent to this choice and can probably be used with other lookback/horizon configurations in future work.

---

> ### Author Response · Authors · 2025-11-21
> **Response to Reviewer Koi8: part 2**
>
> >**(Q1) Please give a comprehensive introduction of your sampled data**
>
> We are happy to clarify how our sampled dataset is constructed.
>  Our data are sampled from the ERA5 reanalysis datasets (details in **Appendix E.1**., page 24), which is defined on a regular $0.25^{\circ} \times 0.25^{\circ}$ lat–lon grid. From this global field, we randomly select four non-overlapping square subregions, each of size $20^{\circ} \times 20^{\circ}$ in latitude and longitude. To avoid the geometric distortion and area shrinkage of high-latitude grid cells on the sphere, we manually exclude all candidates with $|{\rm lat}| > 60^{\circ}$. Under the ERA5 $0.25^{\circ}$ resolution, each selected subregion contains  $81 \times 81$ grid points. On the temporal axis, we construct the dataset at hourly resolution. For each of the four regions, we treat **one day** as one basic sequence unit: each daily sequence consists of 24 consecutive hourly snapshots (i.e., time step $1\text{h}$). Across all four regions, this yields in total of 35064 daily time series, each being a length-24 trajectory on an $81 \times 81$ grid. Before training, we randomly shuffle all daily sequences from all regions together, and then split them into non-overlapping training, validation, and test sets. In this way, all methods are trained and evaluated on exactly the same datasets of spatio-temporal samples, and the reported results focus on relative performance under a common, realistic setting constructed from ERA5.
>
>
> >**(Q2) Did you find an optimal masking threshold to be consistent across different geographic regions and weather regimes, or does it require case-specific tuning?**
>
> We thank the reviewer for raising the importance of frequency masking level as a hyperparameter.
>
> In the current version of the manuscript, we study the effect of the masking threshold in Appendix F.2. As shown in **Figure 5**, sweeping the masking threshold across a wide range yields **very similar trends** for the four regions (EA, EU, NA, AU): the performance curves share a comparable shape and their optima concentrate in a narrow interval. This suggests that the optimal masking level is not highly case-specific to a particular geographic region or weather regime. In our main experiments, we therefore adopt empirically chosen near-optimal thresholds obtained from this sweep, and keep them fixed when reporting the results in Sec. 4.
>
> >**(Q2 cont'd) Is there a potential to make this threshold learnable?**
>
> Regarding the question of making the threshold learnable, we agree that this is a promising extension. In principle, one could treat the masking level as a trainable scalar (or even a small set of region-dependent parameters) and optimize it jointly with the network parameters. More expressive variants could further replace hard masking by a learnable soft mask (for example, perhaps a spatial attention-based weighting) that adaptively emphasizes high-impact regions.
>
> However, such designs also **increase model complexity** and introduce additional parameters that may complicate optimization and interpretation, especially in data-scarce extreme-wind regimes. Therefore, in this paper, we intentionally keep the masking threshold as a simple, empirically selected hyperparameter so that the gains can be more cleanly attributed to the proposed gradient-penalized loss--which is the main contribution of the paper.

---

> > ### Author Response · Authors · 2025-11-27
> > **We look forward to your response as the discussion period ends soon**
> >
> > Dear Reviewer Koi8,
> >
> > Thank you once again for your commitment to reviewing our paper and helping us in improving our work. We would like to remind you that the discussion window will close soon, and we eagerly await your feedback.
> >
> > We have provided detailed explanations for each of your concerns. We would greatly appreciate it if you could review our responses and let us know if they fully or partially address your concerns. Any additional comments you may have would be highly appreciated.
> >
> > Best regards,
> >
> > Authors

---

### Official Review · Reviewer_SvAa · 2025-10-31

**Soundness:** 3
**Presentation:** 3
**Contribution:** 3
**Rating:** 6
**Confidence:** 2

**Summary:**

This paper investigates the underestimation of extreme wind speeds in data-driven weather forecasting models, a persistent issue in both academic research and industrial applications such as wind power management. The authors propose a frequency-aware learning framework that integrates theoretical analysis in the Fourier domain with innovative model and loss function design.

The study first demonstrates through analytical proof that traditional mean squared error training induces frequency-dependent amplitude decay, where high-frequency components are systematically weakened due to spatial translation errors. To address this, the authors propose:
(1) A gradient penalty loss function to enhance sensitivity to amplitude errors and mitigate high-frequency signal attenuation.
(2) A physically embedded neural network architecture: Employing a simplified Navier-Stokes core, it integrates physically grounded convection, diffusion, and pressure modules with a learnable “volume force” network.
(3) Band separation and weight rebalancing mechanism: Input signals are decomposed into high- and low-frequency components, processed separately via Fourier filters, and equipped with time-domain attention mechanisms for each band.

Experiments on the ERA5 meteorological reanalysis dataset demonstrate that this model consistently outperforms CNN, ConvLSTM, and PINN benchmark models in RMSE metrics, achieving significantly improved accuracy in extreme wind regions. Analysis further confirms the stabilizing effect of gradient penalties and the optimal trade-off relationship regulated by their coefficient λ.

**Strengths:**

(1) Strong theoretical insight and motivation: The paper provides a clear Fourier-domain error decomposition explaining why high-frequency components are underestimated by standard MSE. The analytical backing is mathematically sound and bridges physical intuition and machine-learning loss design, a commendable improvement over purely empirical approaches in weather forecasting.
(2) Innovative frequency-informed hybrid framework: The integration of gradient-penalized loss, Navier–Stokes-based backbone, and frequency-domain reweighting demonstrates an elegant blend of physics knowledge and deep learning. The model architecture and spectral treatment are well-motivated, providing an interpretable mechanism for addressing extreme-event underestimation.
(3) Comprehensive experimental design and convincing results: Clear comparisons with strong baselines such as CNN, ConvLSTM and PINN. Analysis of the λ hyperparameter and frequency masking ablations demonstrates methodological robustness. Results on both overall and extreme-attentive errors substantiate the theoretical claims.

**Weaknesses:**

(1) Limited experimental diversity and scale: The study relies on ERA5 data from selected regions, with no cross-region or cross-time validations to assess generalization ability. Suggestion: Extend evaluations to multiple climate zones and temporal ranges (e.g., 6h, 48h forecasts) to assess robustness and transferability of the frequency-informed model.
(2) Computational overhead and implementation detail gaps: The physics-embedded backbone and frequency separation introduce heavy computation during both training and inference. Quantitative data on training time, convergence speed, or complexity trade-offs are missing. Suggestion: Provide complexity analysis versus baseline models, and discuss deployment feasibility for operational forecasting.
(3) Limited connection between frequency-domain theory and physical interpretability: The gradient-penalized term and frequency reweighting are theoretically justified, but it remains unclear how these modifications alter learned spectra or physical consistency over time.
Suggestion: Include frequency-spectrum visualizations pre- and post-training, or energy distribution comparisons to ground truth, to confirm the mitigation of amplitude shrinkage empirically.

**Questions:**

（1）How does the proposed framework perform under longer-term forecasts or coarse-resolution settings where statistical noise dominates over high-frequency content?
（2）Can the gradient-penalized loss lead to overfitting sharp gradients or instability in turbulent regions, and how is λ chosen or adapted dynamically during training?
（3）How generalizable is the physics-embedded structure? Could it extend effectively to 3D atmospheric models or other variables (temperature, humidity) without major redesigns?

---

> ### Author Response · Authors · 2025-11-21
> **Response to Reviewer SvAa: part 1**
>
> We sincerely thank you for your in-depth and thoughtful comments. We greatly appreciate your recognition of the novelty and theoretical insight of our work. We are here to provide detailed replies to your comments and hope we can resolve your major concerns.
>
> ---
>
> >**(W1) Limited experimental diversity and scale: The study relies on ERA5 data from selected regions, with no cross-region or cross-time validations to assess generalization ability.**
>
> >**(Q1) How does the proposed framework perform under longer-term forecasts or coarse-resolution settings where statistical noise dominates over high-frequency content?**
>
> We thank the reviewer for pointing out the need to better assess generalization across regions and forecasting horizons. Following your suggestion, we have added **preliminary cross-region and cross-time validations** in the revised manuscript.
>
> - **Cross-time (longer horizons).** _(See Sec. 4.2, Table 1 on page 9)_
>
>     Our setting targets short-term regional _nowcasting_ for wind power operations, where 6h–48h ahead forecasts (as you have suggested) are typically beyond the operational decision horizon. To still examine longer-term behavior within a realistic range, we extend the experiments from the original 1 h lead time to **3 h and 5 h**. We evaluate all baselines and our model under these horizons, reporting both RMSE and Ex-RMSE. The results are shown as follow and can be found in Table 1 of the manuscript.
>
> | Model    | 1h RMSE    | 1h Ex-RMSE | 3h RMSE    | 3h Ex-RMSE | 5h RMSE    | 5h Ex-RMSE |
> | -------- | ---------- | ---------- | ---------- | ---------- | ---------- | ---------- |
> | CNN      | 0.4639     | 0.3183     | 1.0442     | 0.7355     | 2.0757     | 1.0693     |
> | ConvLSTM | 0.3471     | 0.2294     | 0.7834     | 0.5357     | 1.0644     | 0.8097     |
> | PINN     | 0.3946     | 0.2541     | 0.8283     | 0.5646     | 1.1434     | 0.7347     |
> | **Ours** | **0.3287** | **0.1868** | **0.6622** | **0.4329** | **0.9076** | **0.6158** |
>
> ---
>
> - **Cross-region (OOD) evaluation.** _(See Appendix F.3, Fig. 6 on page 28 and Table 3 on page 29.)_
>
>     Beyond the training regions, we now perform **out-of-distribution tests on three additional regional domains** that are randomly selected from **East Asia (EA)**, **North America (NA)**, and **Australia (AU)**. All models are evaluated under the same setup as in the main experiments. We present the results in the following table:
>
> | Region | Model    | 1h RMSE    | 1h Ex-RMSE | 3h RMSE    | 3h Ex-RMSE | 5h RMSE    | 5h Ex-RMSE |
> | ------ | -------- | ---------- | ---------- | ---------- | ---------- | ---------- | ---------- |
> | EA     | CNN      | 0.9559     | 0.6011     | 3.3106     | 2.2872     | 5.1008     | 3.7931     |
> | EA     | ConvLSTM | 0.8356     | 0.4217     | 2.4391     | 1.5702     | 3.0405     | 2.5460     |
> | EA     | PINN     | 0.8156     | 0.3983     | 2.0649     | 1.4828     | 2.8327     | 2.2321     |
> | EA     | **Ours** | **0.6947** | **0.3236** | **1.8657** | **1.3334** | **2.5840** | **1.9037** |
> | NA     | CNN      | 0.8309     | 0.5691     | 1.9977     | 1.5617     | 3.6052     | 2.4897     |
> | NA     | ConvLSTM | 0.6569     | 0.4237     | 1.2097     | 1.0331     | 1.7852     | 1.5028     |
> | NA     | PINN     | 0.6630     | 0.4863     | 1.3394     | 1.1201     | 1.8592     | 1.6391     |
> | NA     | **Ours** | **0.6065** | **0.3872** | **1.1935** | **0.9305** | **1.6862** | **1.2477** |
> | AU     | CNN      | 1.1043     | 0.6830     | 2.3372     | 1.4968     | 3.0311     | 2.1449     |
> | AU     | ConvLSTM | 0.8584     | 0.3292     | 2.0674     | 1.0791     | 2.4728     | 1.7153     |
> | AU     | PINN     | 0.9342     | 0.4409     | 2.2171     | 1.0831     | 2.3327     | 1.5570     |
> | AU     | **Ours** | **0.7411** | **0.3244** | **1.8853** | **0.8901** | **2.0521** | **1.1371** |

---

> ### Author Response · Authors · 2025-11-21
> **Response to Reviewer SvAa: part 2**
>
> >**(W2) Computational overhead and implementation detail gaps. The physics-embedded backbone and frequency separation introduce heavy computation during both training and inference.”**
>
> We thank the reviewer for raise this point. We now provide a basic complexity discussion of our model and compare it with the reference baselines (here we take ConvLSTM as an example).
>
> Let the spatial grid size be $N\times M$, the number of input/output channels in a convolutional layer be $C_{\rm in}$ and $C_{\rm out}$, and the kernel size be $F\times F$. For one forward pass at a single time step, the complexity of a standard convolution is
> $$
> \mathcal{O}\bigl(N M\, C_{\rm in} C_{\rm out} F^2\bigr).
> $$
> In our implementation, most layers have $C_{\rm in}\approx C_{\rm out} = C$, so a deep ConvLSTM backbone with $L$ layers and $T$ time steps has overall complexity on the order of
> $$
> \mathcal{O}\bigl(T L\, N M\, C^2 F^2\bigr).
> $$
>
> Compared with standard baseline models, the additional computation in our framework mainly comes from three modules: **(i) frequency separation**, **(ii) temporal attention**, and **(iii) the NS-based operator**. Now we discuss them separately.
>
> - **Frequency separation.**
>   We perform one 2D FFT / inverse FFT on each input frame of size $N\times M$. For $T$ input frames, this costs$$
>   \mathcal{O}\bigl(T N M \log(N M)\bigr).
>   $$
>
> - **Temporal attention.**
>   The temporal attention block uses global pooling over space followed by two small fully connected layers on the channel dimension and channel-wise reweighting. If $C$ is the number of feature channels and $r$ the reduction ratio, the additional cost of the temporal attention will be $$
>   \mathcal{O}\bigl(T C + T C^2 / r\bigr),
>   $$
>   which does not scale with $N M$ and is therefore negligible compared to the $\mathcal{O}(T L N M C^2 F^2)$ cost of the host spatial convolution networks.
>
> - **NS-based operator.**
>   The NS block consists of a learnable body-force network plus three simple NS-inspired operators (viscous, pressure-like, and advective terms). In our implementation, **the main cost comes from the body-force network**, whose depth and width are deliberately chosen. For simplicity, here we suppose the body force operator adopts a complexity comparable to the ConvLSTM baselines, i.e. similar order as $\mathcal{O}(T L N M C^2 F^2)$.
>
>   By contrast, the remaining three NS operators are implemented as **pointwise linear combinations and additions** on the 2-channel physical velocity field $(u,v)$. Their cost scales like$$
>   \mathcal{O}\bigl(T N M C_{\rm phy}\bigr), \quad C_{\rm phy}=2,
>   $$
>   which is negligible compared to the convolutional backbone.
>
> Putting these pieces together, the overall time complexity of our model can be written as
> $$\mathcal{O}\left( \max\lbrace T L N M C^2 F^2,\; T N M \log(NM) \rbrace \right),$$
> since it is difficult to analytically compare $\log(NM)$ and $L C^2 F^2$ in full generality. In other words, the runtime is dominated by whichever is larger between the convolutional backbone and the FFT-based frequency separation.
>
> In practice, if the grid size $NM$ is moderate, then the FFT term does not dominate by orders of magnitude. In this case, the cost of our full model will remain comparable to that of the ConvLSTM and CNN baselines on the same grid and hardware. So our method does not introduce fundamentally higher computational requirements.
>
> In conclusion, we agree that, for any finite number of training iterations and for each inference pass, our architecture introduces some additional constant-factor computation compared to the baselines. However, there is no free lunch: this moderate overhead is the cost we pay for our main goal—improving the prediction of extreme events.

---

> > ### Comment · Reviewer_SvAa · 2025-11-24
> >
> > Thank you for your responses. The detailed experiments and analysis have answered my questions, and I am willing to maintain my positive score. Best regards.

---

> ### Author Response · Authors · 2025-11-21
> **Response to Reviewer SvAa: part 3**
>
> >**(W3) Limited connection between frequency-domain theory and physical interpretability**
>
> We thank the reviewer for raising the concern. After analyzing the energy property of the loss function, we are excited to show you that there exist **strong theoretic connection** between the proposed gradient-penalized loss (originally derived from frequency domain knowledge) and physical interpretability. We provide comprehensive explanation regarding the connection in Sec. 3.2 (“Energy–Enstrophy Interpretation of the Gradient-Penalized Loss”), and you may find more detailed proofs and interpretation in Appendix C in the updated manuscript.
>
> In fluid dynamics, the **enstrophy** $\mathcal{E}$ is a potential density related to the kinetic energy that corresponds to the dissipation effects in the fluid. As shown by Theorem 1 on page 5, for an incompressible 2D velocity field $\mathbf{u}$, the enstrophy $\mathcal{E}(\mathbf{u})$ can be expressed (up to constants) as the squared $L^2$-norm of the gradient:
> $$\mathcal{E}(\mathbf{u}) = \int _ \Omega |\omega(x,t)|^2 dx = \int _ \Omega \|\nabla \mathbf{u}(x,t)\|^2 dx = \|\nabla \mathbf{u}\| _ {L^2}^2,$$
> under standard assumptions. Thus $\|\nabla \mathbf{u}\|_{L^2}^2$ measures the total strength of rotation and shear, dominated by fronts, shear layers, and small-scale vortices.
>
> We then provide the classical energy balance for incompressible NS (See Theorem 2 on page 6):
> $$\frac{1}{2}\frac{d}{dt}\|\mathbf{u}(t)\| _ {L^2}^2 + \nu \|\nabla \mathbf{u}(t)\| _ {L^2}^2 = \langle \mathbf{F}(t), \mathbf{u}(t)\rangle.$$
> Here the dissipation term $\nu \|\nabla \mathbf{u}(t)\|_{L^2}^2$ is proportional to enstrophy, and hence directly controls the rate of kinetic energy decay. Matching enstrophy between prediction and ground truth therefore constrains the physically admissible rate of energy change over time.
>
> Moreover, we connect this to the frequency-domain picture via the spectral representation of energy and enstrophy:
> $$
> E(t) = \frac{1}{2}\|\mathbf{u}(t)\| _ {L^2}^2
> = \frac{1}{2}\int _ 0^\infty E(k,t) dk,
> $$
> $$
> \mathcal{E}(t)
> = \int _ \Omega \|\nabla \mathbf{u}(x,t)\|^2 dx
> = \int _ 0^\infty k^2 E(k,t) dk.
> $$
> Both quantities are generated by the same spectrum $E(k,t)$, but enstrophy uses a $k^2$-weighted integral and is therefore heavily biased toward high wave numbers (small scales). This shows that the enstrophy-matching part of $\mathcal{L}_{gp}$ is equivalent to a frequency-weighted matching of spectra, where high-$k$ modes are emphasized in a way that is directly tied to NS energy dissipation.
>
> Based on the above interpretation, our gradient-penalized loss $\mathcal{L}_{gp}$ can be viewed as an explicit **energy–enstrophy trade-off**:
> $$ \mathcal{L} _ {gp}(\tilde{\mathbf{u}},\mathbf{u}) =\underbrace{\int _ \Omega \|\tilde{\mathbf{u}}(x)-\mathbf{u}(x)\|^2 dx} _ {\text{energy matching (MSE)}} + \lambda \underbrace{\big|\|\nabla \tilde{\mathbf{u}}\| _ {L^2}^2- \|\nabla \mathbf{u}\| _ {L^2}^2\big|} _ {\text{enstrophy matching}}.$$
> Here, the first term minimizes the kinetic energy of the error field, while the second term matches the enstrophy, which controls the temporal dissipation rate of kinetic energy. Consequently, the gradient-penalized loss does not only suppress the error energy, but also constrains the predicted flow to exhibit a physically consistent rate of energy change. In practice, the learned model thus maintains an overall accurate prediction of the velocity field, while restoring sufficient gradient and vorticity strength in high-impact regions so that the total enstrophy remains comparable to the ground truth. Therefore, the gradient-penalized loss makes uniform amplitude shrinkage an inefficient way to reduce the objective, encouraging the network to preserve the magnitude of physically relevant small-scale structures.

---

> ### Author Response · Authors · 2025-11-21
> **Response to Reviewer SvAa: part 4**
>
> >**(Q2) Can the gradient-penalized loss lead to overfitting sharp gradients or instability in turbulent regions, and how is λ chosen or adapted dynamically during training?**
>
> We thank the reviewer for raising this important question about potential instability and overfitting of sharp gradients. We agree that, in principle, an excessively large gradient weight λ can make the optimization unstable, especially in highly turbulent (high-enstrophy) regions. To better understand this effect, we provide a convergence analysis of the gradient-penalized loss in **Appendix D on page 23**.
>
> We study a simplified spectrum–wise model of the gradient-penalized objective. For each spatial frequency mode with wave number vector $k$, the effective curvature of the per-spectrum quadratic depends on the factor $(1 - \lambda \|k\|^2)$. The key conclusion is that **stability requires** $\lambda \|k\|^2 < 1$ for all spectrum $k$ with major energies; otherwise, the effective curvature becomes non-positive and there exists no learning rate η such that the standard stability condition
> $0 < \eta (1 - \lambda \|k\|^2) < 1$ can hold. This yields a **theoretical upper bound** on λ of the form
> $$
> \lambda < \lambda _ {\max} \approx \frac{1}{\|K\| _ {\max}^2},
> $$
> where $\|K\|_{\max}$ is the largest spectrum whose dynamics we want to control. Empirically, the majority energy should locate on the spectra with magnitude less than $\|k\| _ {\max}$. Intuitively, if λ goes beyond this range, high-frequency (turbulent) cases can indeed become unstable under gradient descent. The above results is consistent with the U-shape curve shown in **Figure 2 on page 8**.
>
> In our experiments, we do not use a dynamically increasing λ, but instead select a fixed λ within the theoretically stable regime. Concretely, we sweep λ on a validation set and choose the value that minimizes the validation error (as shown in Figure. 2). Empirically, we do not observe exploding gradients or spurious oscillations in turbulent regions with this proper choice.
>
> We agree that region- or scale-adaptive choices of λ might be a promising direction to further reduce the risk of overfitting local sharp gradients while still enforcing physical consistency where needed. Exploring such adaptive λ schemes might be an interesting avenue for future work beyond the scope of this paper.
>
> ---
>
> >**(Q3) How generalizable is the physics-embedded structure? Could it extend effectively to 3D atmospheric models or other variables (temperature, humidity) without major redesigns?**
>
> We thank the reviewer for raising the question.
>
> **On extension to 3D atmospheric models.**
> Conceptually, the NS-based backbone in Sec. 3.3 is not limited to 2D. Changing from 2D to 3D structure would increase memory and computation demands, but does not require a fundamental redesign of the architecture.
>
> However, theoretically, there might be some differences between the properties of 2D&3D NS equations. For example, the energy of 3D NS flow may easily explode for critically small scales, which might require further regularization for the energy. Therefore, we are not sure about the performance of analogous model in 3D cases. But we do believe that it will be an interesting topic for future works to better predict the turbulence in 3D flows.
>
> **On other variables (temperature, humidity, etc.).**
> Our current model is explicitly designed for, but not limited to predicting the wind velocity field. In principle, this formulation is compatible with **any** flow field whose dynamics can be reasonably modeled or approximated by incompressible NS equations, such as ocean surface currents or general numerical fluid simulation.
>
> However, we do **not** claim that this architecture can, without modification, predict all meteorological variables, since many of them are governed by other evolution equations (e.g., thermodynamic or microphysical equations). But if the meteorological variables are strongly driven or affected by the underlying wind field, our approaches might also be applied with certain modificatioins. For example, scalar fields such as precipitation fields or cloud cover can often be modeled as tracers transported by the wind; in such cases, one might extend the backbone by adding extra channels and corresponding advection–diffusion operators.
>
> Designing a universal architecture that jointly handles multiple atmospheric variables under a unified multi-equation framework might be interesting for future work on extreme weather prediction. Therefore, our current design could be viewed as one step in that direction rather than a complete solution.

---

> ### Author Response · Authors · 2025-11-27
> **Thanks for your response!**
>
> Dear Reviewer SvAa,
>
> We are very excited to know that our responses have addressed your concerns, and we are truly grateful for your positive evaluation of our work. Your comments are both constructive and detailed, and we sincerely appreciate the responsibility and thoughtfulness you devoted to reviewing our paper.
>
> If you have any further questions or suggestions that you feel could help us improve the manuscript, please feel free to let us know.
>
> Best regards,
>
> Authors

---

### Author Response · Authors · 2025-11-21
**General response**

We would like to express our sincere gratitude to all reviewers for their constructive comments and valuable suggestions. We have carefully considered each point raised and conducted further theoretical analysis and experiments. We update the corresponding revisions to the manuscript, with the main changes are summarized below:

1. **Theoretical analysis of the gradient-penalized loss.**

    We provide a PDE-based energy–enstrophy interpretation of the proposed loss function. We show that the gradient regularization can be essentially viewed as enstrophy matching, and leads to an explicit energy–enstrophy trade-off that is consistent with the frequency-domain knowledge. This highlights a beautiful alignment between PDE dynamics and signal-processing.

    _(See Sec. 3.2, page 5; and Appendix C, page 16)_

2. **Convergence analysis.**

    We add a convergence analysis for a simplified version of our gradient-penalized optimization dynamics in the Fourier domain, and show that the resulting bounds are consistent with the empirical performance trends as a function of the regularization weight $\lambda$.

    _(See Appendix D, page 23)_

3. **Multi-horizon forecasting experiments.**

    We include experiments over multiple forecasting horizons (including lead times of 1 h, 3 h, and 5 h) for all baseline models and our method.

    _(See Sec. 4.2, page 8, Table 1 on page 9; Appendix F.3, Table 3 on page 29.)_

4. **Cross-regional evaluation.**

    In addition to the training region, we conduct out-of-distribution tests on three randomly selected regional domains from East Asia (EA), North America (NA), and Australia (AU).

    _(See Appendix F.3, Fig. 6 on page 28 and Table 3 on page 29.)_

5. **Ablation study of model components.**

    We add an ablation study that separately removes (i) the frequency-separation module, (ii) the NS-based operator, and (iii) the gradient-penalized loss to examine the role of each component of our model.

    _(See Sec. 4.3, Table 2, page 10)_


We hope that our responses have addressed all the concerns from the reviewers. If there are any further concerns or questions, we are happy to address them and provide further clarifications during the discussion period.

---

### Meta-Review · Area_Chair_yNWj · 2026-01-02

**Summary:**

The authors propose a frequency-informed learning scheme that improves extreme windspeed prediction.
Reviewers generally agree that the paper presents a meaningful advance in extreme weather prediction, which is motivated by empirical observations and theoretical insights regarding existing schemes and their shortcomings.
Reviewers find the theoretical insights in the study and the proposed frequency-informed prediction framework to be novel and interesting, and note the performance evaluations convincingly demonstrate the advantages of the proposed method.
Some initial concerns include that the evaluations are based on limited experimental settings, the manuscript lacks a detailed discussion on the computational overhead, and the need for additional ablation studies and further theoretical insights.

**Reviewer Concerns:**

The authors have thoroughly addressed the reviewers' major concerns during the rebuttal period.
Further evaluations have been performed for additional time horizons (forecast in 1h, 3h, 5h) as well as cross-region predictions.
Ablation studies have been carried out to investigate the contributions of individual model components
The authors also provided additional discussions regarding the convergence and computational overhead.

**Reviewer Scores:**

Two reviewers confirmed that the authors have sufficiently addressed their concerns and opted to maintain their original positive score.
While reviewer Koi8 did not respond to the authors, the authors' response seemed to be adequate, and the reviewer might have either increased or maintained the initial positive score.

---

### Decision · Program_Chairs · 2026-01-26

Accept (Poster)